# Arctic Ocean sea ice cover during the penultimate glacial and the last interglacial

Ruediger Stein[1,2], Kirsten Fahl [1], Paul Gierz[1], Frank Niessen[1] & Gerrit Lohmann[1,3]

Coinciding with global warming, Arctic sea ice has rapidly decreased during the last four decades and climate scenarios suggest that sea ice may completely disappear during summer within the next about 50–100 years. Here we produce Arctic sea ice biomarker proxy records for the penultimate glacial (Marine Isotope Stage 6) and the subsequent last interglacial (Marine Isotope Stage 5e). The latter is a time interval when the high latitudes were significantly warmer than today. We document that even under such warmer climate conditions, sea ice existed in the central Arctic Ocean during summer, whereas sea ice was significantly reduced along the Barents Sea continental margin influenced by Atlantic Water inflow. Our proxy reconstruction of the last interglacial sea ice cover is supported by climate simulations, although some proxy data/model inconsistencies still exist. During late Marine Isotope Stage 6, polynya-type conditions occurred off the major ice sheets along the northern Barents and East Siberian continental margins, contradicting a giant Marine Isotope Stage 6 ice shelf that covered the entire Arctic Ocean.

[1] Alfred Wegener Institute Helmholtz Centre for Polar und Marine Research (AWI), Am Alten Hafen 26, 27568 Bremerhaven, Germany. [2] Department of Geosciences (FB5), University of Bremen, Klagenfurter Str. 4, 28359 Bremen, Germany. [3] MARUM-Center for Marine Environmental Sciences, University of Bremen, Leobener Straße, 28359 Bremen, Germany. Correspondence and requests for materials should be addressed to R.S. (email: Ruediger.Stein@awi.de)

Sea ice with its strong seasonal and interannual variability (Fig. 1) is a very critical component of the Arctic system that responds sensitively to changes in atmospheric circulation, incoming radiation, atmospheric and oceanic heat fluxes, as well as the hydrological cycle[1, 2]. Ice significantly reduces the heat flux between ocean and atmosphere; through its high albedo it has a strong influence on the radiation budget of the entire Arctic. Furthermore, the sea-ice cover strongly affects biological productivity, as a more closed sea-ice cover reduces primary production due to low light influx in the surface waters. As sea ice is sensitive to atmospheric and oceanic variability, and because it is involved in several key climate feedbacks (ice-albedo feedback, cloud-radiation feedback, etc.), sea ice plays a substantial role in the global climate system variability, known as polar amplification[3].

Over the past three to four decades, coincident with global warming and atmospheric $CO_2$ increase, Arctic sea ice has significantly decreased in its extent (cf., Fig. 1) as well as in thickness[2, 4–6]. The loss of sea ice results in a distinct decrease in albedo, causing further warming of ocean surface waters. When extrapolating this trend, the central Arctic Ocean might become ice-free during summers within about the next five decades or even sooner[2, 7]. Based on a proxy reconstruction, ice-free summers also occurred during a late Miocene warm climate with simulated atmospheric $CO_2$ concentrations of 450 ppm[8], a value we also might reach in the near future. Furthermore, the recent decrease in sea ice seems to be more rapid than predicted by climate models[4, 5], indicating that the processes causing these recent rapid climate changes are not fully understood and subject of intense scientific and societal debate. In this context, a key aspect is to distinguish and quantify more precisely natural and anthropogenic greenhouse gas forcing of global climate change and related sea ice decrease[2].

The last time that Arctic temperatures were significantly higher than today was the Early Holocene Thermal Maximum[9, 10]. The Holocene, however, is an interglacial cycle not concluded yet. This certainly justifies climatic evaluations of older, concluded warm interglacial cycles such as the last interglacial (LIG), i.e., Marine Isotope Stage (MIS) 5e (Eemian), lasting from about 130 to 115 ka and often proposed as a possible analog for our near-future climatic conditions on Earth[11, 12]. Based on proxy records from ice, terrestrial and marine archives, the LIG is characterized by an atmospheric $CO_2$ concentration of about 290 ppm, i.e., similar to the pre-industrial (PI) value[13], mean air temperatures in Northeast Siberia that were about 9 °C higher than today[14], air temperatures above the Greenland NEEM ice core site of about 8 ± 4 °C above the mean of the past millennium[15], North Atlantic sea-surface temperatures of about 2 °C higher than the modern (PI) temperatures[12, 16], and a global sea level 5–9 m above the present sea level[17]. In the Nordic Seas, on the other hand, the Eemian might have been cooler than the Holocene due to a reduction in the northward flow of Atlantic surface water towards Fram Strait and the Arctic Ocean, indicating the complexity of the interglacial climate system and its evolution in the northern high latitudes[12, 18, 19].

If climate models are able to reproduce past warm climatic conditions (such as those of the LIG), including the extent of Arctic sea ice cover, we will have additional confidence in their representation of Arctic processes and their projections for the future[20–23]. In order to test and approve climate models for simulation and prediction of Arctic climate and sea ice cover[8, 20–28], however, precise (semi-quantitative) proxy records about past sea ice concentrations are needed. Such records may be obtained using a quite recently developed biomarker approach that is based on the determination of a highly branched isoprenoid (HBI) with 25 carbons ($C_{25}$ HBI monoene)[29]. This biomarker is only biosynthesized by specific diatoms living in the Arctic sea ice and thus named 'IP$_{25}$' (= ice proxy with 25 carbons)[30]. That means, the presence of IP$_{25}$ in the sediments is a direct proof for the presence of past Arctic sea ice. Meanwhile, this biomarker approach has been used successfully in many studies dealing with the reconstruction of Arctic sea ice history during the last glacial to Holocene time interval, i.e., the last about 30 ka[31–37]. Furthermore, this biomarker seems to be quite stable as it was found in sediments as old as the late Miocene[8]. By combination of this sea ice proxy IP$_{25}$ with (biomarker) proxies for open-water phytoplankton productivity such as brassicasterol, dinosterol or a specific tri-unsaturated HBI (HBI-III)[37–41], a more precise (semi-quantitative) reconstruction of present and past Arctic Ocean sea ice conditions from marine sediments are now available (Supplementary Fig. 1; see Metho 6ds for some more details). For older glacial and interglacial intervals such as MIS 6 and MIS 5, however, no such biomarker data of the central Arctic Ocean sea ice cover are available so far. For these time intervals, reconstructions of past sea ice conditions are mainly restricted to continental margin sites and, even more important, only based on indirect proxies such as, for example, foraminifera, dinoflagellates, and ostracodes[42–48].

Here we produce these biomarker proxy records of sea ice distribution in the central Arctic Ocean for the time interval of late MIS 6–MIS 5. Including open-water phytoplankton biomarkers as well as micropaleontological data, we demonstrate (1) that a permanent sea ice cover existed during MIS 6 and (2) that during the LIG sea ice was still present in the central Arctic Ocean during the spring/summer season even under (global) boundary conditions significantly warmer than the present. Seasonal open-water conditions, on the other hand, occurred along the Barents Sea continental margin during the interstadials of MIS 5 (with minimum values during MIS 5e/Eemian) but also during the preceding glacial MIS 6. The latter finding—although still based on a low-resolution record—appears to contradict the hypothesis of a thick ice shelf covering the entire Arctic Ocean during MIS 6 as proposed by Jakobsson et al.[49]. Our proxy records are compared with climate model simulations using a coupled atmosphere-ocean general circulation model.

## Results

**Glacial to LIG Arctic Ocean sea ice cover.** In order to reconstruct the sea ice history of the Arctic Ocean during glacial (MIS 6) to LIG (MIS 5) conditions, we determined the sea ice biomarker proxy IP$_{25}$, open-water phytoplankton biomarkers, and terrestrial biomarkers from four selected sediment cores (see Supplementary Table 1 for exact core locations and water depths, Supplementary Tables 2–5 for the biomarker data). These cores were recovered from areas characterized by different sea ice conditions today, ranging from perennial sea ice in the central Arctic Ocean to seasonal sea ice conditions along the Barents Sea continental margin (Fig. 1; see Methods for more details).

Both IP$_{25}$ as well as brassicasterol concentrations are zero or close to zero throughout the time interval from MIS 6 to MIS 5 in the two high-Arctic cores PS2200-5 and PS51/038-3 (Fig. 2a, b). In all studied samples from both cores also no HBI-III was found (Supplementary Tables 2 and 3). These data strongly point to predominantly perennial sea ice cover during the glacial and LIG (Fig. 3b), preventing algal production during the spring and summer. Based on the biomarker data, the MIS 6/MIS 5 sea ice conditions in the central Arctic Ocean were probably similar to those reconstructed for the Last Glacial Maximum (LGM) and MIS 1/Holocene (Fig. 3a, Supplementary Fig. 6)[36, 40].

The biomarker records of the MIS 6/MIS 5 interval at Core PS2757-8 display a surprising distribution pattern (Fig. 2c). Maximum concentrations of sea ice, open-water phytoplankton

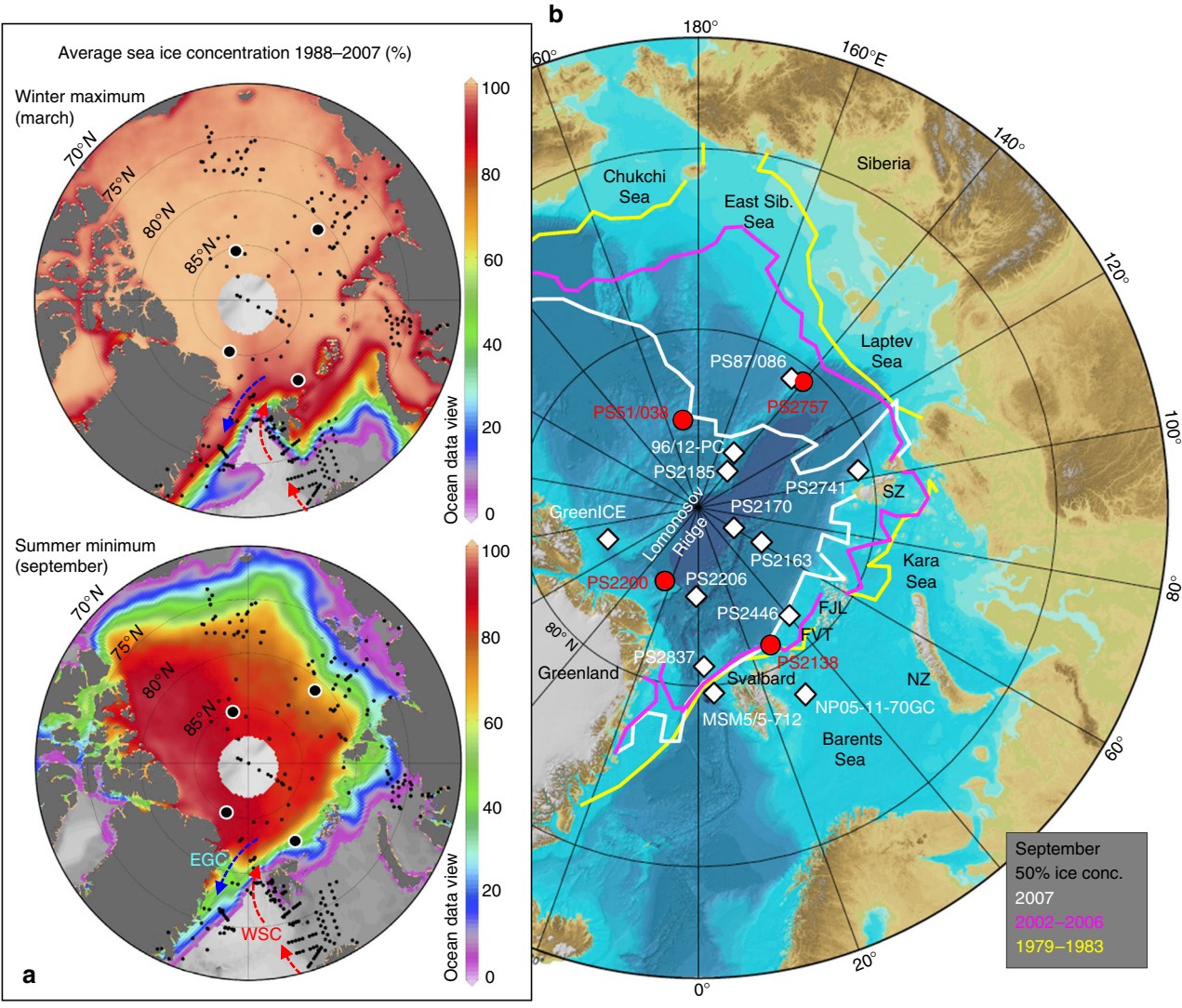

**Fig. 1** Modern Arctic sea ice concentrations and location of sediments cores discussed in this study. **a** Average sea ice concentration 1988–2007 for March (winter maximum) and September (summer minimum) (Source: http://nsidc.org/). Small *black dots* show locations of surface sediment samples used for the IP$_{25}$ synthesis study of modern sea ice distribution[40], the four large black circles show locations of the key cores used in this study. EGC=East Greenland Current; WSC=West Spitsbergen Current. Maps produced by software 'Ocean Data View'[75]. **b** IBCAO bathymetry map of the Arctic Ocean[76] with locations of sediments cores used in this study (cf., Supplementary Table 1). *Red circles*=key cores of this study; *white rhombs*=cores used for discussion. NZ = Novaya Zemlya; SZ = Severnaya Zemlya; FJL = Franz Josef Land; FVT = Franz Victoria Trough. The *white, pink,* and *yellow lines* show September 50% sea ice concentration for the years 2007, 2002–2007, and 1979–1983, respectively (source: http://iup.physik.uni-bremen.de)

and terrestrial plant biomarker proxies, i.e., IP$_{25}$, brassicasterol and HBI-III, as well as ß-sitosterol, respectively, are recorded during late MIS 6, followed by a sharp drop during Termination II at the end of MIS 6. During MIS 5 including the LIG, on the other hand, all biomarker proxies display minimum values of zero or close to zero (except for one IP$_{25}$ peak near the base of MIS 5). Based on these data and looking at the position of the data points within the IP$_{25}$ vs. brassicasterol diagram (Fig. 3b), an extended sea ice cover with occasional ice edge/polynya conditions likely prevailed in the southern Lomonosov Ridge area close to the Siberian continental margin during the late MIS 6. Under such conditions, ice melting and related nutrient and sediment release may have resulted in high fluxes of ice algae, open-water phytoplankton and terrigenous matter (Supplementary Fig. 1). In contrast, a more or less closed sea ice cover probably existed during MIS 5.

Whereas at the three sites from the central Arctic Ocean an extended to perennial sea ice cover was probably predominant,

sea ice conditions were much more variable along the northern Barents Sea continental margin as reflected in the data of Core PS2138-2 (Fig. 2d). High concentrations of both the sea ice proxy IP$_{25}$ and the phytoplankton biomarker brassicasterol were measured in samples representing the late MIS 6 and the interstadials MIS 5e, 5c and 5a. For the latter interstadials, also increased HBI-III concentrations were determined (see discussion below). Based on these data, an extended but variable sea ice cover with closed sea ice to ice-edge conditions occurred during late MIS 6 (Fig. 3b, Supplementary Fig. 1). During the MIS 5 interstadials, a seasonal sea ice cover and ice-edge conditions seem to have been most prominent, with minimum sea ice concentrations towards almost ice-free summers during MIS 5e (Eemian) (Fig. 3b). During the stadials MIS 5d and 5b, on the other hand, sea ice and phytoplankton biomarkers display zero to almost zero concentrations, indicative of a more closed sea ice cover (Figs. 2d and 3b).

**Last interglacial Arctic Ocean sea ice cover simulations.** In addition to our proxy records, we performed transient integrations as well as time slice experiments for the MIS 5 (see Methods). The model experiments were driven by orbitally-induced insolation and greenhouse gas concentrations in the atmosphere (Supplementary Table 6). Figure 4 shows the time slices for 130, 125, and 120 ka as well as the PI conditions. Our simulation efforts reveal similar boreal spring (March) sea ice extent in the LIG time slices as in the PI simulation. While sea ice concentrations were slightly lower in June during the early LIG (130 ka) and the middle LIG (125 ka) compared to PI concentrations, the largest difference can be observed in September when sea ice concentrations during the early and middle LIG were distinctly lower then those modeled for the PI. During the late LIG (120 ka), on the other hand, September sea ice concentrations seems to have been quite similar to the PI (Fig. 4).

## Discussion

The Quaternary glacial history of the Arctic Ocean is characterized by the repeated build-up and decay of circum-Arctic ice sheets on the continental shelves, the development and disintegration of ice shelves, and related changes in ocean-circulation patterns and sea ice cover[50–55]. There is, however, still an ongoing and partly controversial debate about the timing and extent of maximum glaciations. A comprehensive circumpolar overview of glacial landforms, stratigraphies, and chronologies and their interpretation in terms of glacial history, is given by Jakobsson et al.[54], summarizing the current state of knowledge and identifying key questions arising from this synthesis. Based on new evidence of ice-shelf groundings on bathymetric highs in the central Arctic Ocean, Jakobsson et al.[49] most recently proposed an extended thick ice shelf covering the entire central Arctic Ocean (Fig. 5b) and dated it to MIS 6 (~140 ka).

This hypothesis would be in line with the biomarker data from the central Arctic Ocean sites PS2200-5 and PS51/038-3 pointing to a more closed and thick ice cover that has prevented both phytoplankton as well as sea ice algae production (Figs. 2a, b, 3b). The near absence of planktic foraminifers in the MIS 6 sediments of these cores (Supplementary Figs. 2 and 3)[56] also supports the interpretation of virtually no surface water productivity due to closed sea ice conditions. From these data, however, it would not be possible to distinguish between a closed, several meters thick sea ice cover and an extended ice shelf of several hundred of

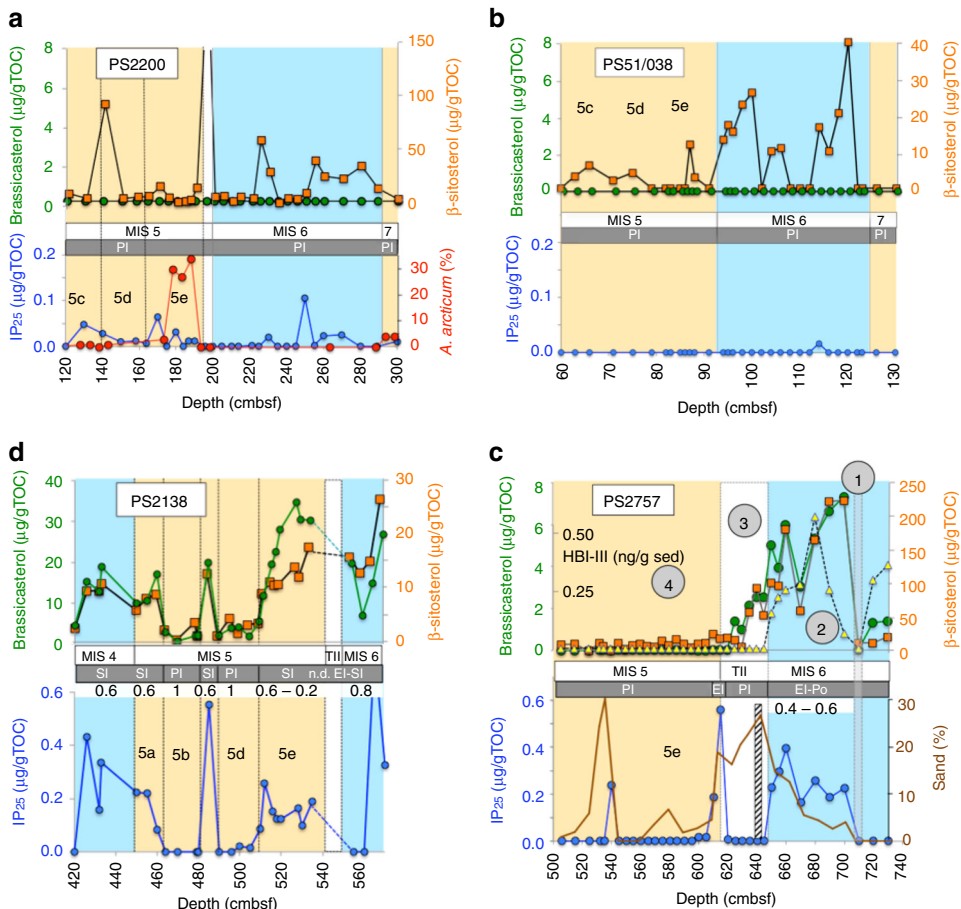

**Fig. 2** Proxy records for sea ice and surface-water productivity at the four studied sites for the time interval MIS 6—MIS 5. **a** Core PS2200-5, **b** Core PS51/38-3, **c** Core PS2757-8, and **d** Core PS2138-2. Concentrations (in µg/gTOC) of biomarkers IP25 (*blue circles*), brassicasterol (*green circles*) and ß-sitosterol (*orange squares*) are plotted vs. core depth in centimetres below seafloor (cmbsf). For Core PS2757-8, concentrations (in ng/g sediment) of the HBI-III are also shown (*yellow triangles*). Interglacial MIS 5 (and 7) and glacial MIS 6 (and 4) are highlighted by *beige* and *light blue background color*, respectively. PI = perennial sea ice cover; EI = extended sea ice; EI-Po = extended sea ice and polynya situation. Numbers (0.2, 0.2–0.6, and 1) on top of the IP25 records of cores PS2757-8 and PS2138-2 indicate mean PIP25 index values. For Core PS2200-5, relative abundance of ostracode species *Acetabulastoma arcticum*, indicative for perennial sea ice cover in the central Arctic Ocean[46], are shown (cf., Supplementary Fig. 8). For Core PS2757-8, the amount of sand (*brown solid line*, mainly representing terrigenous material) and a peak event of IRD input (*hatched bar* at 640 cmbf) are added (data from ref. [77]). The *circled numbers* 1–4 indicate different stages of sea ice and ice sheet extent presented in Fig. 6

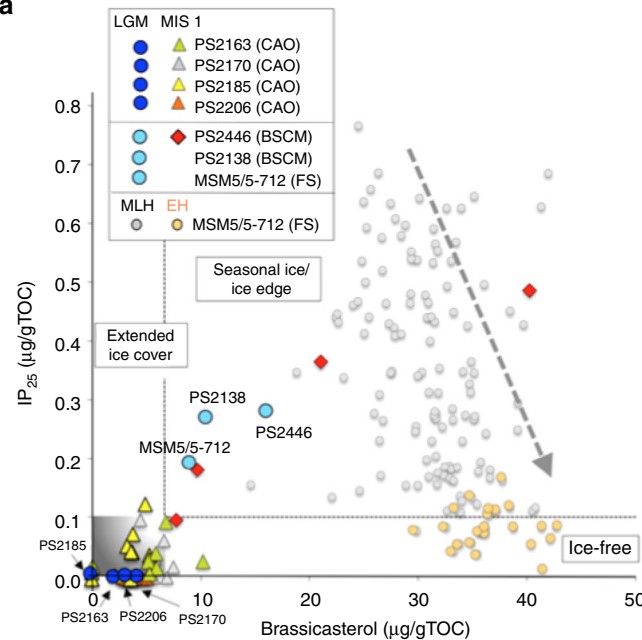

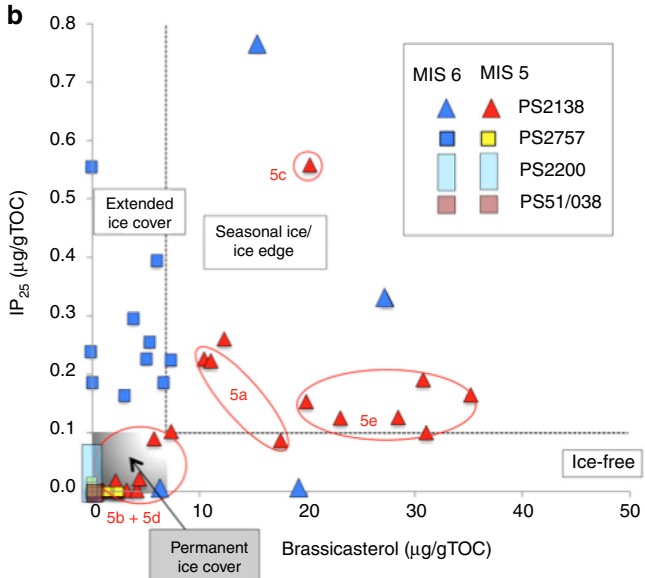

**Fig. 3** IP$_{25}$ vs. brassicasterol plot, indicative for sea ice conditions. Based on this plot, a classification of sea ice conditions into permanent sea ice, extended sea ice, seasonal sea ice (ice edge), and ice-free is possible. The gray box close to the origin marks "permanent/perennial sea ice conditions" characterized by zero or near-zero concentrations of IP$_{25}$ and brassicasterol. **a** Last Glacial Maximum (*LGM*) and MIS 1 conditions for selected cores from the central Arctic Ocean (*CAO*), the Barents Sea continental margin (*BSCM*) and Fram Strait (*FS*)[36] and Early Holocene (*EH*) and Middle-Late Holocene (*MLH*) data from Fram Strait Core MSM5/5-712[32]). **b** Data from cores PS2200-5, PS51/38-3, PS2757-8, and PS2138-2 representing the MIS 6 and MIS 5 time intervals (this study). For MIS 5 of Core PS2138-2, 5a, 5b, 5c, 5d, and 5e data are shown separately

meters in thickness. The data from cores PS2757-8 and PS2138-2 (Figs. 2c, d, 3b), on the other hand, do not support the hypothesis of such a giant late MIS 6 ice shelf. Our biomarker proxy records indicate at least occasionally open-water conditions, i.e., an ice edge situation, that allowed phytoplankton and ice algae production as well as increased flux of terrigenous matter (Fig. 5a; cf., Supplementary Fig. 1).

For the Barents Sea continental margin (i.e., the location of Core PS2138-2), the MIS 6 situation might have been similar to that of the LGM[57, 58]. These authors postulated an extended Barents Sea Ice Sheet, the western part of the huge Eurasian Ice Sheet[51, 55], that had reached the shelf edge causing polynya-like open-water conditions (triggered by strong katabatic winds) with phytoplankton and sea ice algae production, subglacial meltwater outflow and the deposition of suspended material on the slope at site PS2138-2. Furthermore, the seasonal open-water conditions along the Barents Sea continental margin might have been fostered by the inflow of Atlantic Water (Fig. 5a) that was probably significantly reduced but still penetrated continuously to at least the Franz Victoria Trough west of Franz Josef Land (see Fig. 1 for location) during the last 150 ka[58]. Further towards the east, i.e., off Severnaya Zemlya, no clear signals for open-water conditions have been found in the sedimentary record of Core PS2741-1 (Fig. 5a)[58].

But how can the open-water conditions at Core PS2757-8, i.e., the southern Lomonosov Ridge close to the East Siberian continental margin not influenced by Atlantic Water during MIS 6, be explained? One probable explanation could be that an extended East Siberian Chukchi Ice Sheet (ESCIS) as proposed by Niessen et al.[53] has existed at this time (Fig. 5a). Evidence for glacial landforms based on hydro-acoustic data from the East Siberian continental margin remain undated[53], but were most recently supported by numerical modeling to be related to an ESCIS that has formed during MIS 6[59]. Such an extended ice sheet associated with strong katabatic winds should have caused polynya-like open-water conditions in front of the ice sheet (Fig. 5a), resulting in increased fluxes of phytoplankton, ice algae and terrigenous matter as observed in the PS2757-8 record (Fig. 2c and Fig. 6, Scenario 2), i.e., a situation similar to that proposed for the Barents Sea continental margin (Fig. 5a)[57, 58]. Such a scenario would clearly contradict the hypothesis of a MIS 6 ice shelf covering the entire Arctic Ocean (Fig. 5b)[49]. On the other hand, Core PS2757-8 is close to the area where we discovered SE-NW oriented streamlined landforms over distances of >100 km on top of the southern Lomonosov Ridge at water depths between 800 and 1000 m during the Polarstern Expedition PS87, interpreted as glacial lineations that were formed by coherent masses of grounded ice flowing across the ridge in a NW direction (Supplementary Fig. 7d)[8]. Based on sub-bottom profiling and the age model of Core PS87/086-3, a MIS 6 age of the youngest ice-erosional event seems to be most realistic (Supplementary Figs. 4, 7a, 7b). This observation would support the hypothesis of an extended MIS 6 ice shelf at least for this area.

One possible but still somewhat speculative explanation for these discrepancies could be a temporal succession of different scenarios as shown in Fig. 6. After a period with maximum extension of the ESCIS covering the southern Lomonosov Ridge (including the area of cores PS2757-8 and PS87/086-3) and causing ice-shelf grounding (ice rise) with no ice algae production underneath (Fig. 6, Scenario 1), the ice shelf started to retreat. Polynya-like conditions caused by strong katabatic winds allowed sea ice algae and phytoplankton production during the late(st) MIS 6 (Fig. 6, Scenario 2). A marginal ice zone (MIZ)/polynya situation may also be supported by the presence of low but significant concentrations of HBI-III (Fig. 2c). Although the exact sources of this biomarker are not known yet, it seems to be strongly enhanced in MIZ environments as described for the Antarctic[60] as well as the Arctic (Fig. 7f)[37]. During Termination II (Fig. 6, Scenario 3), the ESCIS collapsed close to the shelf edge, and numerous icebergs calved near the grounding line of the remaining ESCIS of which some drifted over the coring location as shown in peak abundances of the terrigenous sand fraction and peak accumulation of ice-rafted debris (IRD) (Fig. 2c). During the

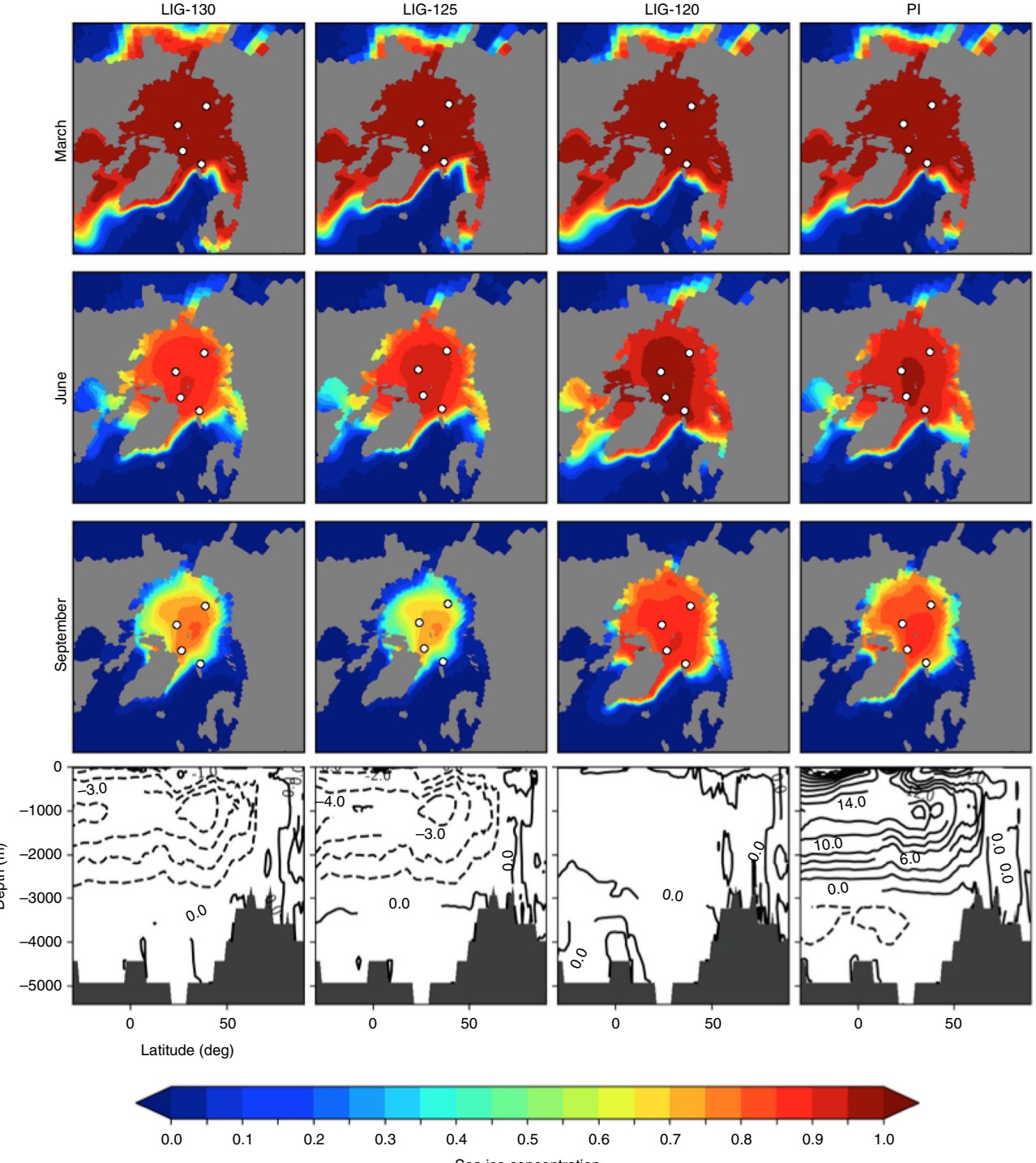

**Fig. 4** Simulation of Arctic sea ice cover of the Last Interglacial and the pre-industrial climate. Last Interglacial (*LIG*) conditions were simulated for three time slices: LIG-130 (130 ka), LIG-125 (125 ka), and LIG-120 (120 ka). White circles indicate locations of the four studied sediment cores. The Atlantic Meridional Overturning Circulation (*AMOC*) is shown in the lower panels of Fig. 4 for each time slice. For the PI, numbers indicate absolute Sv values whereas the numbers in the LIG runs are Sv anomalies relative to the PI values (i.e., -3.0 means a reduction of three Sv in comparison to the PI value). The AMOC during LIG-130 and LIG-125 is reduced compared to the Pre-Industrial (*PI*) control runs. The increase in AMOC at 120 ka relative to the 125 ka leads to an increased oceanic heat transport and partly compensates for the decrease in insolation

following LIG (MIS 5e/Eemian), the ESCIS disappeared (and with it the katabatic winds) and an extended, more or less closed sea ice cover remained over the southern Lomonosov Ridge in the area of Core PS2757-8, preventing phytoplankton and sea ice algae productivity (Fig. 6, Scenario 4). Towards the flooded East Siberian shelf, it is likely that ice-free conditions existed. The

interpretation of the biomarker records of Core PS2757-8 is also further supported by some limited biomarker data from nearby Core PS87/086-3, representing the scenarios 1, 2, and 4 described in Fig. 6 (Supplementary Figs. 4 and 7).

As described above for the biomarker proxies of Core PS2200-5 and Core PS51/038-3 (Figs. 2 and 3), a perennial sea ice cover

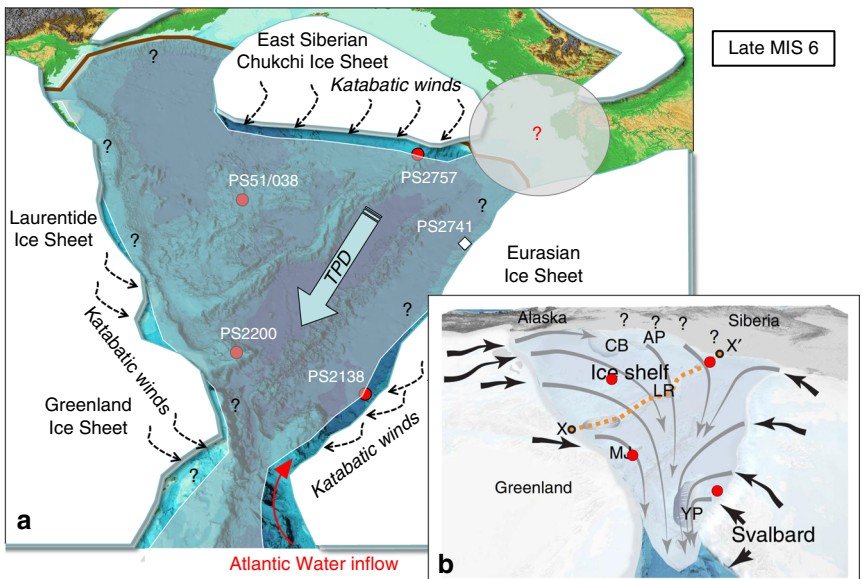

**Fig. 5** Schematic illustration of Arctic sea ice cover and circum-Arctic ice sheets during MIS 6. **a** Ice sheet configuration[52, 54], including the extended ice sheet on the East Siberian continental margin[53]. Such an extended East Siberian ice sheet/shelf seems to be supported by numerical reconstruction, but should have been connected with the Eurasian Ice Sheet (transparent light *gray oval* with *red question mark*[59]). Strong katabatic winds related to the ice sheets (shown tentatively as *stippled black arrows*), were probably responsible for ice-free polynya-type conditions off the major ice sheets, causing phytoplankton and sea-ice algae productivity recorded in cores PS2138-3 and PS2757-8 (for the region off the Greenland-Laurentide Ice Sheet no proof from sediment cores are available. Thus, a polynya-type situation is only assumed and marked by *question marks*). Large *light blue arrow* indicates the Transpolar Drift (*TPD*). **b** Cartoon of an ice shelf covering the entire central Arctic Ocean with flow lines generalized from mapped glacial landforms[49]. AP, Arlis Plateau; CB, Chukchi Borderland; LR, Lomonosov Ridge; MJ, Morris Jesup Rise; YP, Yermak Plateau; x–x′ highlights the crest of the Lomonosov Ridge. In **a** and **b**, locations of the four studied key cores are shown as *red circles*

was also predominant in the central Arctic Ocean during MIS 5, including the LIG (MIS 5e/Eemian), i.e., a period that was probably significantly warmer than the present (Holocene)[12, 14, 15]. Planktic foraminifers (predominantly the cold-water species *Neogloboquadrina pachyderma* sin.), however, are present during the LIG at both sites PS2200-5 and PS51/038-3 (Supplementary Figs. 2 and 3)[56]. Furthermore, the total amounts of planktic foraminifers are very similar to those determined in Holocene sediments from these two cores (Supplementary Figs. 2 and 3), suggesting similar sea ice conditions during the LIG as during the latest Holocene (present). That means, the perennial sea ice cover must have been interrupted by phases with some restricted open-water conditions during summer that allowed foraminifers to reproduce[56]. The latter is also supported by the presence of calcareous algae (coccolithophoridae) in the Eemian sediments of Core PS2200-5 (Supplementary Fig. 2)[56]. Furthermore, this interpretation is in line with high abundances of the ostracode species *Acetabulastoma arcticum* found in these sediments of Core PS2200-5 (Fig. 2a) and Core 96/12-1PC from Lomonosov Ridge (see Fig. 1 for location), proposed to be a proxy for a perennial sea ice cover with >75% sea ice concentrations (Supplementary Fig. 8)[46]. However, phytoplankton- and ice algae-related organic matter production and flux must have been very low. Thus, it did not (Core PS51/038-3) or almost not (Core PS2200-5) survive zooplankton grazing, transport through the water column and degradation at the seafloor and in the sediment.

Peak abundances of the small subpolar planktic foraminifer species *Turborotalita quinqueloba* found in MIS 5e sediments from the southern Lomonosov Ridge close to the Greenland continental margin (Site GreenICE, Fig. 1), a region with a modern perennial sea ice cover, may indicate less sea ice than today[45]. According to these authors, however, it cannot be determined whether a reduction in sea ice cover

was part of a more wide-spread regional pattern or a more restricted phenomenon forced by a polynya-type setting (similar to the modern NorthEast Water Polynya off northeast Greenland).

Along the Barents Sea continental margin, the sea ice cover was quite variable throughout the entire MIS 5. This variability in sea ice cover was probably mainly triggered by the variability in Atlantic Water inflow[42, 56, 58] and (at least in the early-mid MIS 5e) driven by solar insolation (Fig. 7). During stadials MIS 5d and 5b, the inflow of Atlantic water was strongly weakened, indicated by the near absence of the dinoflagellate species *Operculodinium centrocarpum* (Fig. 7d)[42] and resulting in a strongly extended sea ice cover. Such an extended sea ice cover and reduced primary production are reflected in the near absence to absence of both $IP_{25}$ (Fig. 7b) and phytoplankton biomarkers (i.e., brassicasterol and HBI-III) (Fig. 7c) as well as maximum $PIP_{25}$ values (Fig. 7a). This interpretation is further supported by the minimum of the total number of dinoflagellate cysts and peak concentrations of the dinoflagellate species *Impagidinium pallidum* (Fig. 7d), indicative of cold polar conditions and an extensive seasonal sea ice cover[42]. During the interstadials and coinciding with maxima in insolation, on the other hand, sea ice was reduced and surface-water productivity increased as indicated by minima in $PIP_{25}$ and peak values in brassicasterol and HBI-III, respectively (Fig. 7a, c). Especially the latter may may point to increased productivity in connection with a MIZ situation[37, 60].

The most prominent sea ice minimum occurred during the LIG (MIS 5e/Eemian) as clearly reflected in the semi-quantitative $PIP_{25}$ records of Core PS2138-2. Both $P_BIP_{25}$ and $P_{III}IP_{25}$ (see Methods for further explanation) reach very similar minimum numbers of about 0.2 and less (Fig. 7a), i.e., numbers that may correspond to spring/summer sea-ice concentration of about 20% or even less (cf.,[37, 38]). We relate this sea ice minimum to the strong inflow of warm Atlantic Water as indicated by the

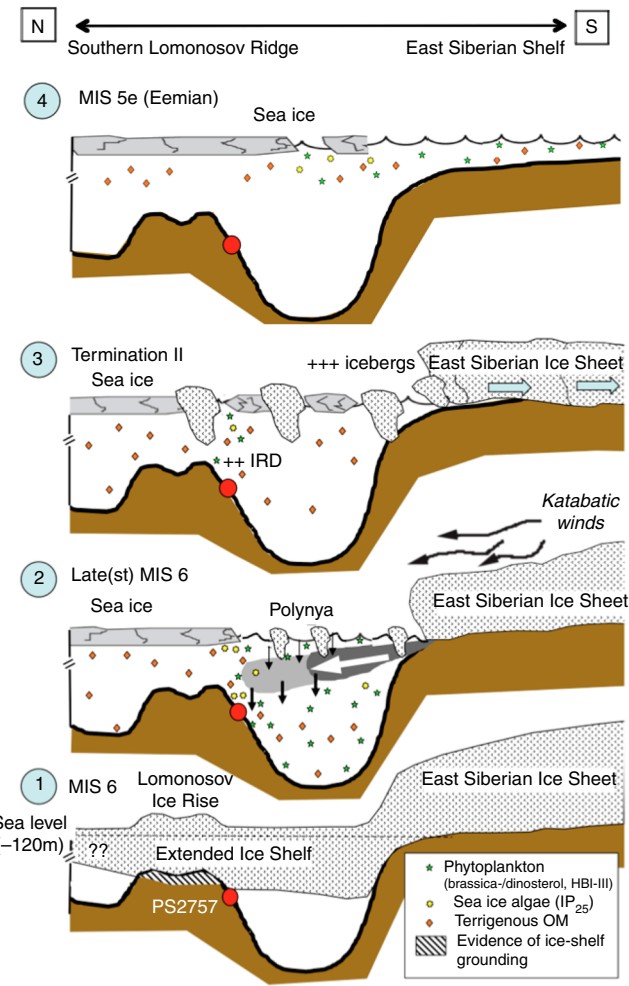

**Fig. 6** Cartoon of different paleoenvironmental scenarios of sea ice and ice sheet extent at the East Siberian Continental Margin/Southern Lomonosov Ridge area. These scenarios are (1) to (4) based on the biomarker records from Core PS2757-8 (Fig. 2c). (1) MIS 6 scenario with an extended East Siberian Ice Sheet and ice shelf/rise. Under such conditions, neither phytoplankton productivity nor terrigenous organic matter (*OM*) input occur as reflected in the absence of the biomarkers, and total sedimentation may have decreased to zero. Hatched field marks area on southern Lomonosov Ridge where traces of erosion by grounding ice were recorded as SE-NW oriented streamlined bedforms on the Lomonosov Ridge at water depths between 800 and 1000 m (ref. [8]; cf., Supplementary Fig. 7). (2) Late MIS 6 with the ice sheet still reaching the shelf edge, with a polynya situation caused by strong katabatic winds. Quite stable ice-edge conditions resulted in increased fluxes of IP$_{25}$, open-water phytoplankton and terrigenous biomarkers (cf., Supplementary Fig. 1). (3) Phase of major retreat and decay of the ice sheet resulting in high sediment (IRD) input by calving icebergs. (4) Last Interglacial (MIS 5e/Eemian) with a more or less closed sea ice cover situation over Core PS2757-8, preventing phytoplankton and sea ice algae productivity, and probably ice-free conditions towards the East Siberian shelf. The occurrence of phytoplankton, sea ice algae and terrigenous biomarkers are indicated by *green stars*, *yellow stars* and *orange rhombs*, respectively. *Red circle* indicates approximate location of Core PS2757-8

contemporaneous prominent maxima in open-water phytoplankton biomarker concentrations (i.e., brassicasterol and HBI-III), total numbers of dinoflagellate cysts and—especially—the dinoflagellate species *O. centrocarpum* (Fig. 7d) as well as a prominent maximum in Atlantic-Water species of benthic foraminifers[44]. This interval of strongly reduced summer sea ice

concentration shows some delay to the summer insolation maximum (Fig. 7b), but is more or less contemporaneous with a peak LIG warmth documented in the Nordic Seas[61–64]. Unfortunately, our time resolution of one sample/3000 years is not high enough to distinguish between the early, middle and late LIG conditions as it is possible for the data sets from the Nordic Seas. There, planktic δ[18]O records from cores MD95-2010 and MD99-2304 (for core locations see Fig. 7e) document a climatic optimum in the early-middle part of the LIG between about 126 and 116 ka, related to a strong poleward extension of warm Atlantic Water[61, 62, 64]. These conditions are quite similar to those also described for the Early Holocene at cores MSM5/5-712-2 and NP05-11-70GC (Fig. 7a, g; see Fig. 7e for core locations), i.e., very low PIP$_{25}$ values of 0.2 and less, interpreted as almost ice-free conditions triggered by increased Atlantic Water inflow[37, 38].

Towards the end of the LIG, i.e., during the early stage of the last glacial inception after 115 ka and coinciding with a minimum of the northern summer insolation, sea ice along the Barents Sea continental margin started to extend by a factor of 2–3 in comparison to the mid-LIG/Eemian minimum (Fig. 7a, b). Contemporaneously, the central Arctic Ocean sea ice concentrations probably increased as well (cf., Fig. 2a). Based on simulation experiments, a related increase in Arctic freshwater export by sea ice may have induced a weakening in ocean heat transport by the subpolar gyre in the North Atlantic[20].

Whereas most proxy-based reconstructions point to an early-middle LIG climatic optimum with reduced summer sea ice concentrations between 126 and 116 ka, the results of our model simulations only support a pronounced reduction in summer sea ice concentration for the LIG-125 and LIG-130 runs (in both time slice as well as transient runs; Figs. 8 and 9), but also indicate that sea ice was still present in the central Arctic Ocean even under climatic conditions significantly warmer than today (Fig. 4). The presence of summer sea ice in the central Arctic Ocean—despite the elevated air temperatures[14, 15, 64]—may have resulted from a reduced total oceanic heat flux towards the north, as suggested from (compared to the PI control runs) reduced AMOC patterns during LIG-130 and LIG-125 (Fig. 4). In the central Arctic Ocean, simulated LIG-130 and LIG-125 sea ice concentrations decreased to about 65–75% at sites PS51/038-3 and PS2757-8 (Figs. 8b, c) and to about 50–60% at site PS2200-2 (Fig. 8a). At the Barents Sea continental margin (i.e., at site PS2138-2) strongly influenced by Atlantic Water inflow, minimum summer sea ice concentrations of about 25% were simulated for the 125 ka time slice (Fig. 8d). This minimum value fits almost exactly with the PIP$_{25}$ reconstruction at Core PS2138-2 (Fig. 7a).

For the LIG-120 interval, we record an apparent mismatch between the LIG-120 simulation (suggesting sea ice conditions similar to those of the PI conditions) (Figs. 4 and 8) and the proxy-based sea ice record (suggesting minimum sea ice concentrations similar to the early-mid-LIG (Fig. 7a). A similar mismatch between LIG-120 mean annual surface temperature (MAT) simulation and proxy data is also described by Otto-Bliesner et al.[21]. The simulated MAT is very similar to the PI, whereas the LIG-130 and LIG-125 MAT simulations resulted in significantly warmer temperatures. In our study, this mismatch might be explained by the results of the transient simulations of the LIG from 130 ka to 115 ka, indicating the occasional occurrence of almost ice-free years (probably caused by internal variability) near 120 ka (Fig. 9). During these exceptional years, significantly increased algal productivity may have caused the biomarker signal to be preserved in the sediments. However, this phenomenon cannot explain the mismatch to the other published proxy records clearly indicating warm climatic conditions in the High Northern Latitudes at that time (see discussion above). Thus, there appears to be a need for model

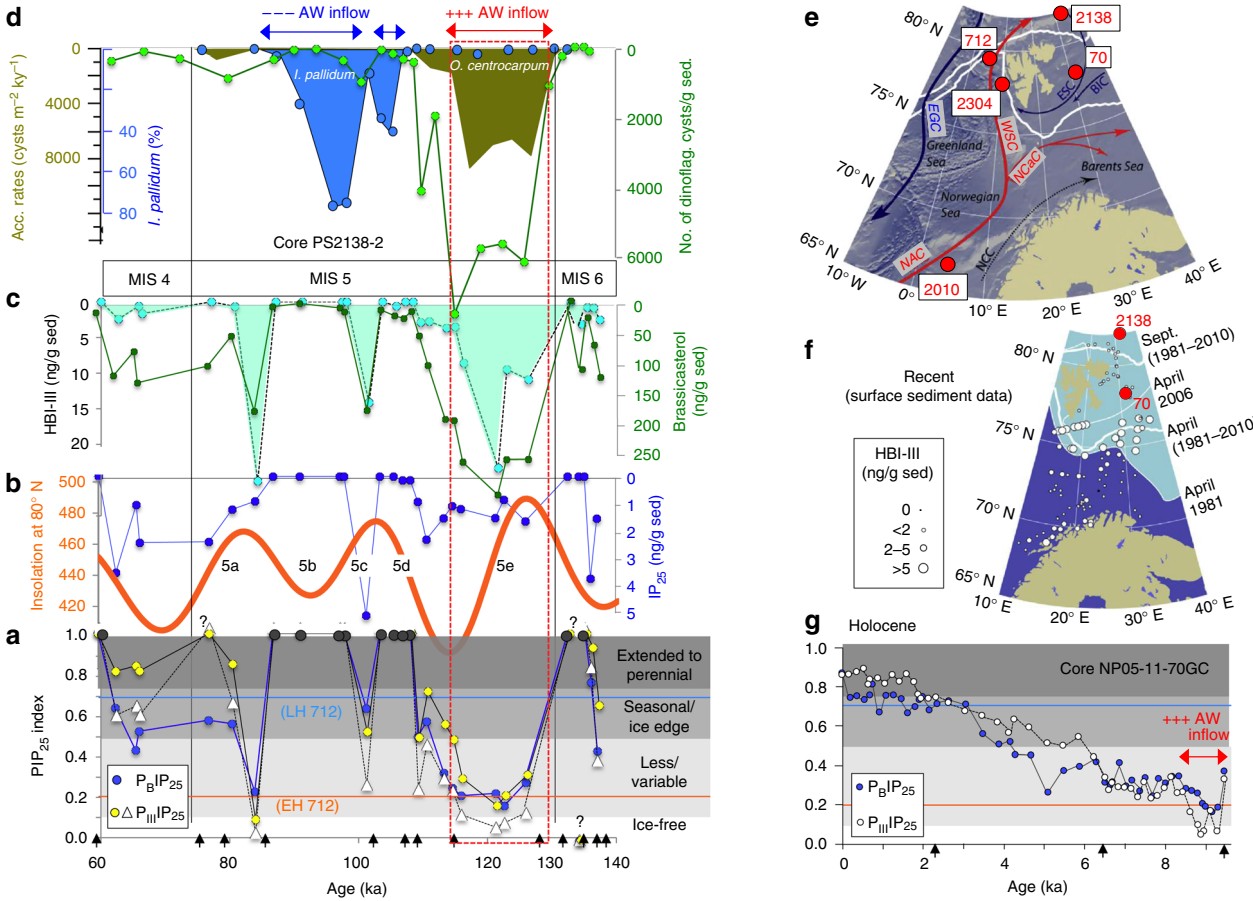

**Fig. 7** Sea ice proxy and dinoflagellate cyst records of the time interval 140–60 ka at Core PS2138-2. The age model of Core PS2138-2 is based on ref. [67].
**a** PIP25 index values representing semi-quantitatve data of sea ice concentrations, based on HBI-III ($P_{III}IP_{25}$; *yellow circles* $c = 0.32$, *white triangles* $c = 1$) and brassicasterol ($P_{B}IP_{25}$; *blue circles* $c = 0.016$); for background and calculation of c factors see Methods. For permanent (perennial) sea ice conditions characterized by zero or near-zero concentrations of $IP_{25}$ and phytoplankton biomarkers (cf., Figs. 2d and 3b), $PIP_{25}$ is indeterminable and set to "1" (*dark gray circles*). $PIP_{25}$ values <0.1, between 0.1 and 0.5, 0.5 and 0.75, and >0.75 point to ice-free conditions, a reduced/variable sea ice cover, a seasonal sea ice cover including an ice edge situation, and an extended to perennial sea ice cover, respectively. EH 712 (*orange horizontal line*) and LH 712 (*blue horizontal line*) indicate mean Early Holocene and Latest Holocene values, respectively[32]. Concentrations in ng/g sediment of **b** $IP_{25}$ and **c** HBI-III and brassicasterol. Summer insolation at 80 °N is shown as an *orange curve*[78]. **d** Numbers of total dinoflagellate cysts per gram sediment (green circles), concentration of sea-ice related dinoflagellate species *Impagidinium pallidum* in percent of total dinoflagellate cyst taxa (*blue circles*) and accumulation rates of *Operculodinium centrocarpum* (*olive-colored graph*) indicative of warm Atlantic Water inflow[42]. *Blue* and *red horizontal arrows* indicate time intervals with reduced and increased Atlantic Water inflow, respectively. **e** Map showing major surface currents (NAC—North Atlantic Current; NCaC—North Cape Current; WSC—West Spitsbergen Current; ESC—East Spitsbergen Current; BIC—Bear Island Current; NCC—Norwegian Coastal Current) and locations of cores PS2138-2, MSM5/5-712, NP05-11-70GC, MD99-2304, and MD95-2010 (from ref. [37], supplemented). **f** Surface sediment concentrations of HBI-III (ng/g sediment) and positions of median April and September sea ice extent (1981–2010), maximum (1981) and minimum (2006) April sea ice extent and locations of cores PS2138-2 and NP05-11-70GC (from ref. [37], supplemented). **g** Holocene $PIP_{25}$ records based on HBI-III ($P_{III}IP_{25}$; *white circles*) and brassicasterol ($P_{B}IP_{25}$; *blue circles*) in Core NP05-11-70GC, with c factors of 1 and 0.01, respectively. *Red horizontal arrow* indicates interval with increased Atlantic Water (AW) inflow, characterized by minimum $PIP_{25}$ values (minimum sea ice extent) (from ref. [37], supplemented). The *black arrows* on the x-axis in **a** and **g** indicate positions of AMS [14]C ages

improvement, which could be solved with further data-modeling comparison.

Pfeiffer and Lohmann[28] indicate in model experiments that the height of the Greenland Ice Sheet also affects the sea ice cover in the Arctic Ocean. That means, a reduced Greenland Ice Sheet would result in a significantly reduced sea ice cover as shown in our LIG-130 simulation (Fig. 8): at the Barents Sea continental margin, sea ice concentrations may have reached values as low as 10% (Site PS2138-2), in the central Arctic Ocean sea ice may have decreased to concentrations of 30–50% (Sites PS51/38-3 and PS2200-2). These very low sea ice concentration values, however, are not supported by our proxy records (cf., Figs. 3 and 7), suggesting that the Greenland Ice Sheet has probably not strongly deviated from its present hight.

Based on ice-volume-equivalent sea level reconstructions, the Bering Strait was probably already re-opened during all the three LIG intervals and most parts of the shallow Siberian marginal seas were already flooded even during LIG-130[65, 66]. As the exact reconstruction of global sea level rise during Termination 2 is still under debate, we also simulate a LIG-130 scenario with a closed Bering Strait and only half-flooded Siberian shelf seas (Supplementary Fig. 9). Whereas the simulations for March and June are all quite similar, the September sea ice concentration of the central Arctic Ocean is significantly lower under conditions with a closed Bering Strait and half-flooded shelf seas (Fig. 8, Supplementary Fig. 9). That means, the simulations with an already re-opened Bering Strait and mostly flooded shelf seas are much more similar to our proxy

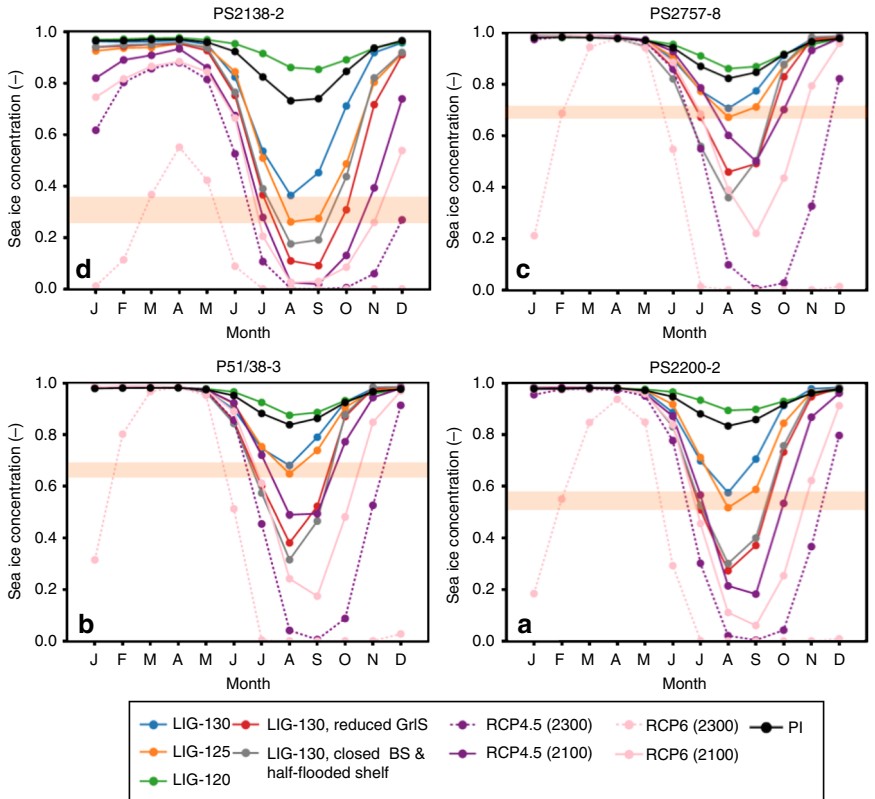

**Fig. 8** Modeled monthly sea ice concentrations at the four different core locations. For **a** Core PS2200-2, **b** Core PS52/38-3, **c** Core PS2757-8, and **d** Core PS2138-2 simulation results for the Last Interglacial (*LIG*) phases at 130, 125, and 120 ka as well as the Pre-Industrial (*PI*) climate are presented. Additionally, we show results from a sensitivity study with a reduced Greenland Ice Sheet (GrIS) at 130 ka[28] and for a 130 ka scenario with a closed Bering Strait (*BS*) and half-flooded Siberian shelf seas (Supplementary Fig. 9) as well as for future (2300 and 2100) scenarios following the IPCC Representative Concentration Pathway (RCP) scenarios RCP4.5 (583 ppm $CO_{2eq}$) and RCP6 (808 ppm $CO_{2eq}$)[2, 27]. Horizontal *orange bars* highlight summer sea ice minima of the LIG-125 and LIG-130 scenarios

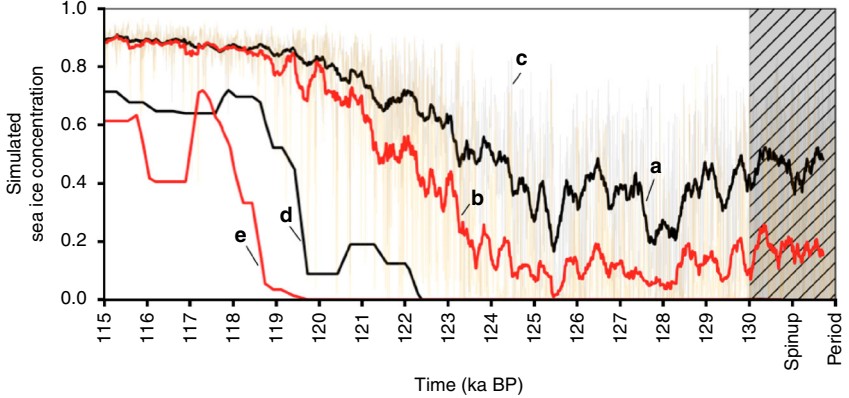

**Fig. 9** Transient simulation of the 130–115 ka time interval, with site location of Core PS2138-2 in *black* (**a**) and the location shifted by one model grid point to the south in *red* (**b**). Semi-transparent time series (**c**) show the yearly simulation output, while the thick lines **a** and **b** indicate a 30-year running average, and a 100-year minimum of the 30-year running average. The last of these time series (running minimum) **d** and **e** demonstrates the signal that might realistically be captured by the biomarker proxy records. That means, at the location of Core PS2138-2 (**d**) sea ice concentrations of <10% might have occasionally occurred at 120 ka

reconstruction, and thus seem to represent the more realistic scenarios.

Finally, we have compared the Arctic sea ice conditions of the LIG and simulated future climate projections for 2100 and 2300, based on two different IPCC scenarios[2], the RCP4.5 (583 ppm $CO_{2eq}$) and the RCP6 (808 ppm $CO_{2eq}$) (Fig. 8). Both scenarios show a severe reduction in sea ice coverage in the late summer,

i.e., summer sea ice concentrations are significantly lower than those of the LIG. With increasing atmospheric $CO_2$, however, the reduction of sea ice in the central Arctic Ocean is more rapid and disproportionately high in comparison to its margin. Whereas the mid-LIG summer sea ice concentrations were still around 60 to 75% in the central Arctic Ocean, but only around 20% or less along the Atlantic-Water influenced Barents Sea continental

margin, nearly ice-free conditions might be reached in the entire Arctic Ocean in 2300. The number of ice-free summer months is increasing with higher atmospheric $CO_2$. Under these high $CO_2$ concentrations, the winter sea ice may start to melt as well (Fig. 8). Furthermore, the higher obliquity during the LIG (Supplementary Table 6) may suggest an insolation forcing during the LIG, whereas for the climate scenarios RCP4.5 and RCP6 the additional heat fluxes are induced by increased greenhouse gas concentrations in the atmosphere.

In conclusion, we are aware that our low-resolution proxy study of the Arctic Ocean sea ice cover during MIS 6 and MIS 5 is only a first but important step. Furthermore, our results also display some model-proxy data inconsistencies (cf.,[21]). Nevertheless, our findings already provide important groundtruthing data of sea ice conditions during the penultimate glacial and the LIG/Eemian of the central Arctic Ocean and the Siberian-Barents Sea continental margin, that may help to further test and improve models for simulation and prediction of future climate change. In follow-up research, such a study should be extended to a high-resolution approach to be carried out on thicker, well-dated MIS 6/MIS 5 sequences recovered from key areas of the Arctic Ocean and adjacent marginal seas. Such sedimentary sequences with the essential spatial and temporal resolution, however, are not available yet and have to be cored in the future. These new sea ice proxy records are needed (1) to fully prove the scenarios of a succession from an extended ice shelf to polynya/open-water conditions (cf., Fig. 6), (2) to reconstruct in more detail the changes in sea ice cover for early, middle and late LIG intervals characterized by very different external forcings and related internal feedback mechanisms, and (3) to allow a more fundamental proxy data/modeling comparison that results in model improvements and better reproduction of the LIG climatic evolution and prediction of future climatic scenarios[20–23, 64].

## Methods

**Studied sediment cores and stratigraphic framework.** Exact locations and water depths of the four studied sediment cores as well as those discussed in the text are listed in Supplementary Table 1. Two of these cores, Core PS2200-5 and Core PS51/038-3, are located in the central Arctic Ocean characterized by a perennial sea ice cover today (Fig. 1; 8–10/10 summer sea ice concentration). Core PS2757-8 is located on the southern Lomonosov Ridge close to the Laptev Sea continental margin, an area that is predominantly covered by sea ice (Fig. 1; 7/10 summer sea ice concentration) but may occasionally be even ice-free during summer. The fourth core, Core PS2138-2, is located at the Barents Sea continental margin, an area with a seasonal sea ice cover and a strong influence of warm Atlantic Water inflow today (Fig. 1; ca. 4/10 summer sea ice concentration).

The stratigraphic framework and related age models of the four sediment cores used in this study, are based on oxygen isotope stratigraphy, [10]Be stratigraphy, paleomagnetostratigraphy, biostratigraphy, lithostratigraphy, and/or magnetic susceptibility records (Supplementary Figs. 2–5). From these data we are confident that the chronology of our records is reliable and accurate enough as a framework for our MIS 6/MIS 5 sea ice reconstruction.

Core PS2200-5: MIS 6 and MIS 5 (5e) have mainly been identified by [10]Be stratigraphy, oxygen and carbon isotope stratigraphy, occurrence of coccolithophoridae, and abundances of planktic foraminifers and coarse fraction (Supplementary Fig. 2)[56] as well as the occurrence of ostracode species[46]. Core PS51/038-3: MIS 6 and MIS 5 (5e) have mainly been identified by oxygen and carbon isotope stratigraphy, paleomagnetostratigraphy, and abundances of planktic foraminifers and coarse fraction (Supplementary Fig. 3)[56]. Core PS2757-8: MIS 6 and MIS 5 (5e) have mainly been identified by lithostratigraphy, paleomagnetostratigraphy, and correlation with other dated sediment cores (Supplementary Fig. 4). Core PS2138-2: MIS 6 and MIS 5 (5e) have mainly been identified by oxygen isotope stratigraphy, paleomagnetostratigraphy, and the occurrence of benthic foraminifera species *Pullenia bulloides* (P.b.) (Supplementary Fig. 5)[58, 67, 68]. In addition, high abundances of benthic foraminifers[44] and dinoflagellate cysts[42, 43] indicate MIS 5e (Fig. 7).

**Biomarker analyses.** Extraction of 5–10 g of freeze-dried sediments was carried out using an accelerated solvent extractor (DIONEX, ASE200; 100 °C, 5 min, 1000 psi) with dichloromethane:methanol (2:1, v/v) as a solvent. For comparison, a set of samples was also extracted by ultra sonic techniques giving similar results (K. Fahl, unpublished data 2017; cf.[69]). For quantification internal standards, 7-

hexylnonadecane (7-HND, 0.076 µg per sample for $IP_{25}$ quantification), squalane (2.4 µg per sample), and cholesterol-$d_6$ (cholest-5-en-3β-ol-$D_6$, 10 µg per sample for sterol quantification), were added prior to analytical treatment. Separation of the hydrocarbon and sterol fractions was carried out via open column chromatography (hydrocarbon fraction with 5 ml $n$-hexane, the sterol fraction with 6 ml $n$-hexane:ethylacetate (5:1, v/v). The latter fraction was silylated with 200 µl bis-trimethylsilyl-trifluoroacet-amide (BSTFA) (60 °C, 2 h). $IP_{25}$, tri-unsaturated HBI (HBI-III, Z-isomere[41]), and sterols were analyzed by gas chromatography (GC) using an Agilent Technologies 7890B (30 m DB-1MS column, 0.25 mm i.d., 0.25 µm film thickness) coupled to an Agilent Technologies 5977A Extractor mass selective detector (MSD, 70 eV constant ionization potential, Scan 50–550 m/z, 1 scan/s, ion source temperature 230 °C, Performance Turbo Pump). GC analyses were performed with the following temperature program for the hydrocarbons: 60 °C (3 min), 150 °C (rate: 15 °C/min), 320 °C (rate: 10 °C/min), 320 °C (15 min isothermal) and for the sterols: 60 °C (2 min), 150 °C (rate: 15 °C/min), 320 °C (rate: 3 °C/min), 320 °C (20 min isothermal). The injection volume was 1 µl splitless. Helium was used as carrier gas (1 ml/min constant flow). Component assignment was based on comparison of GC retention times with those of reference compounds and published mass spectra (cf. Supplementary Fig. 10). The retention indices for brassicasterol (as 24-methylcholesta-5,22E-dien-3β-O-Si($CH_3$)$_3$) and β-sitosterol (as 24-ethylcholest-5-en-3β-O-Si($CH_3$)$_3$) were calculated to be 1.018 and 1.077 (normalized to cholest-5-en-3β-ol-$D_6$ set to be 1.000), respectively.

For the quantification of $IP_{25}$ and HBI-III their molecular ions ($m/z$ 350 and $m/z$ 346, respectively) in relation to the abundant fragment ion $m/z$ 266 of the internal standard (7-HND) were used (selected ion monitoring, SIM mode). The different responses of these ions were balanced by an external calibration[33]. The Kovats Index calculated for $IP_{25}$ is 2086. The detection limit for quantification of $IP_{25}$ using the GC–MS system described is 5 ng m$L^{-1}$. Brassicasterol and β-sitosterol were quantified as trimethylsilyl ethers using the molecular ions $m/z$ 470 and $m/z$ 486, respectively, in relation to the molecular ion $m/z$ 464 of cholesterol-$D_6$.

When using this biomarker proxy for sea ice reconstructions, however, one should have in mind that $IP_{25}$ is absent under a permanent sea ice cover limiting light penetration and, as a consequence, sea ice algal growth (i.e., $IP_{25} = 0$). As $IP_{25}$ is only produced within the sea ice matrix, ice-free conditions also result in zero $IP_{25}$ concentrations. Thus, the two extremes, ice-free vs. thick closed sea ice cover, cannot be distinguished solely by using the $IP_{25}$ biomarker. By considering also a phytoplankton biomarker indicative for open-water primary production, these extremes can be easily separated as under a permanent sea ice cover the phytoplankton biomarker is absent but reaches maximum concentrations under open-water conditions (Fig. 3, Supplementary Fig. 1)[31, 38]. For more semi-quantitative estimates of present and past sea-ice coverage, Müller et al.[38] combined the sea-ice proxy $IP_{25}$ and a phytoplankton biomarker and calculated a phytoplankton-$IP_{25}$ index, the so-called 'PIP$_{25}$ index':

$$PIP\ 25 = [IP_{25}]/([IP_{25}] + ([\text{phytoplankton marker}] \times c)) \qquad (1)$$

with the balance factor $c$ = mean $IP_{25}$ concentration/mean phytoplankton biomarker concentration for a specific data set or core. As open-water phytoplankton biomarkers brassicasterol and dinosterol were used in this approach (see also ref.[39] for further references and critical discussion of these sterols as organic source indicators). The balance factor $c$ is needed due to the significant concentration difference between $IP_{25}$ and brassicasterol (or dinosterol). For the calculation of the mean concentration values of a specific core (interval) zero concentrations should be excluded. The coupling of $IP_{25}$ with phytoplankton biomarkers such as brassicasterol or dinosterol proves to be a viable approach to determine (spring/summer) sea ice conditions as is demonstrated by the good alignment of the PIP$_{25}$-based estimate of the recent sea ice coverage with satellite observations[38]. More recently, Smik et al.[41] introduced the HBI-III alkene as a phytoplankton biomarker replacing the sterols in the PIP$_{25}$ calculation. This modified PIP$_{25}$ approach is far less dependent on the balance factor $c$ and based on biomarkers from the same group of compounds (i.e., HBIs) with more similar diagenetic sensitivity. In a study of sediment cores from the western Barents Sea and the northern Norwegian Sea, Belt et al.[37] have calculated PIP$_{25}$ values using brassicasterol ('P$_B$IP$_{25}$') and HBI-III ('P$_{III}$IP$_{25}$') as a phytoplankton biomarker, respectively. Importantly, these authors could demonstrate that both approaches yielded similar outcomes if the core-specific balance factors were used (Fig. 7g), a fact also supported by our own data here. In this study, we have followed the P$_B$IP$_{25}$ as well as P$_{III}$IP$_{25}$ approaches for sediments from Core PS2138-2, using balance factors of $c$ = 0.016 for P$_B$IP$_{25}$ and both $c$ = 1 and $c$ = 0.32 for P$_{III}$IP$_{25}$ (Fig. 7a; see Supplementary Table 5).

**Model simulation.** The simulations of the Arctic sea ice condition during the LIG are performed with the climate model COSMOS, which consists of the atmosphere model ECHAM5, the ocean/sea-ice model MPI-OM, and the land surface model JSBACH[70]. ECHAM5 and JSBACH both run on a resolution of T31, corresponding to ~3.75° × 3.75° laterally, and the atmosphere is divided in 19 unevenly spaced layers. The ocean model offers a spatial resolution of 1.5° × 3.5° and is vertically divided into 40 unevenly spaced layers, with model poles placed over Greenland and Antarctica. Sea ice formation and dynamics are simulated in the ocean model.

We simulate four distinct periods, three from the LIG corresponding to LIG-130 (130 ka), LIG-125 (125 ka), and LIG-120 (120 ka) with greenhouse gas and orbital values derived from ice core records[71–73] and astronomical calculations[74]; see Supplementary Table 6 for model forcings. As a fourth control period, we simulate the PI conditions. In addition, we also have run a scenario with a closed Bering Strait and half-flooded shelf seas (Supplementary Fig. 9). All of these model experiments were spun up for 2000 years to equilibrate to the changed boundary conditions. Following the equilibration period, we simulate an additional 100 years to be used for evaluation. The simulation with reduced Greenland ice sheet coverage is discussed in more detail in Pfeiffer and Lohmann[28]. Furthermore, we simulate the evolution of the LIG transiently, varying both greenhouse gas values and orbital values as the simulation progresses. This simulation is accelerated by a factor of 10, meaning that every simulated year accounts for 10 years of actual time, and the forcing is correspondingly also accelerated. LIG-130, LIG-125, and LIG-120 are compared to the periods in our transient simulation, and the resulting 100-year means over these time slices are nearly identical to the time slice experiments. The future scenario integrations follow the protocol in the IPCC Representative Concentration Pathway (RCP) scenarios RCP4.5 (583 ppm $CO_{2eq}$) and RCP6 (808 ppm $CO_{2eq}$)[2, 27].

**Data availability**. All data generated or analyzed within this study are included in this published article (and its supplementary information files) and available at https://doi.org/10.1594/PANGAEA.874357.

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

## Acknowledgements

This publication is a contribution to the Research Programme PACES II, Topic 3 (The earth system from a polar perspective: Data, modeling and synthesis) of the Alfred Wegener Institute Helmholtz Centre for Polar und Marine Research (AWI). The study used samples and data provided by AWI (Grant No. AWI-PS87_01). Thanks to Simon Belt (Biogeochemistry Research Centre, University of Plymouth/UK) for providing the 7-HND standard for IP$_{25}$ quantification. We thank three anonymous reviewers for numerous constructive suggestions for improvement of the manuscript.

## Author contributions

R.S. developed the concept of the study and wrote the main manuscript. K.F. conducted the biomarker analyses, evaluation, and quality control. P.G. and G.L. developed the model and experimental design, P.G. carried out the experiments. F.N. compiled the Parasound data. All authors contributed to the data interpretation, graphics and writing process of the final version of the manuscript.

## Additional information

**Competing interests:** The authors declare no competing financial interests.

