## [Peer Review File. · Nature Communications]

Reviewers' comments:

Reviewer #1 (Remarks to the Author):

This paper presents Arctic sea ice reconstructions based on semi-quantitative lipid biomarkers from four sediment cores for a time period (late MIS 6 to MIS 5, including MIS 5e) for which such reconstructions have to date not been available. During MIS 5e, or the Eemian, the Arctic was significantly warmer than today, serving as an analogue for the current rapid warming in the High North. The four study sites are well chosen representing different environmental settings, with two sites located in the central Arctic Ocean, the third close to the Laptev Sea continental margin and the fourth site at the Barents Sea continental margin. The study includes highly important findings based on the biomarker data and supported by microfossil data from previous work: 1) During MIS 6, polynya-like conditions most likely occurred off the major circum-Arctic ice sheets, contradicting a previously hypothesised giant MIS 6 ice shelf that would have covered the entire Arctic Ocean. 2) Even during the Eemian, when climate was clearly warmer than today, sea ice existed in the central Arctic Ocean during summer, whereas ice was significantly reduced along the Barents Sea continental margin influenced by Atlantic Water inflow.

The manuscript is generally well-written and clear. I would, however, suggest checking the English carefully once more, as there were mistakes and also somewhat clumsy sentences (some of these corrected in the pdf of the ms).

The main conclusions of the paper are well-supported by the data. This work is definitely of interest to a wide audience in several fields and of an importance and novelty that warrants publication in Nature Communications.

I have added my comments directly to the manuscript. There is one matter in particular I would like the authors to address, namely adding – if at all possible – data on HBI III. I suspect the authors will have analysed this biomarker alongside IP25. Although the nature of this biomarker proxy is not yet fully understood, there are strong indications it could be used as a marginal ice zone (MIZ) / polynya indicator. See for example: 1) Collins et al 2013. Evaluating highly branched isoprenoid (HBI) biomarkers as a novel Antarctic sea-ice proxy in deep ocean glacial age sediments. 2) Belt et al. 2015. Identification of paleo Arctic winter sea ice limits and the marginal ice zone: Optimised biomarker-based reconstructions of late Quaternary Arctic sea ice. The use of such a proxy would certainly further strengthen the interpretations made in the Stein et al. manuscript. Belt et al. 2015 also point out that the use of HBI III instead of a phytoplankton marker (usually brassicasterol) when calculating the PIP25 –index could produce more reliable results as this index does not potentially suffer from the issue of a variable balance factor (more on this in my specific comments added directly onto the manuscript). The work could be reproduced based on the level of detail provided.

Please note also the figures include comments.

Reviewer #2 (Remarks to the Author):

Comments on: Arctic Ocean sea ice cover during the penultimate glacial and last interglacial

The study by Stein et al explores the environmental conditions of the Arctic Ocean during the last interglacial period (Eemian) and its preceding glaciation phase (stage 6). Their strategy is to use new data of sea-ice biomarker indices to make assumptions on the existence of sea-ice and open-ocean ice shelf coverage for these two time intervals. Their motivation probably stems from other studies that claim for a thick ice shelf covering the entire AO basin during stage 6, and an almost sea-ice free summer during the warm Eemian interglacial period. The proxy data are then backed

up by some modelling efforts for 3 Eemian time slices for different seasons.

Overall, this is an interesting topic and some of the data are indeed quite unique/new by comparison to previous studies from the Arctic Ocean. Also, the main messages contained in the abstract are timely, thus, the story has potential to be published in Nat Comm eventually.

Having said, the implications of the interpretation of the proxy data remains somewhat shallow, in particular, there is little use being made of the model results leaving the entire story rather underdeveloped. I think most of it owes to the fact that the text strays away, is not really focussed on those 2 critical messages talked about in the abstract. In many respect the manuscript has a strong review character reiterating stratigraphic and other data that have been discussed and published at length elsewhere. It starts with an overly long introduction, but continues like that throughout the rest of the manuscript. The discussion part remains by and large descriptive...

Some further comments:

There are quite a few inconsistencies or loose expressions used; some selected examples to follow....

For instance, line 32, 'sea-cover restricts primary productivity....' Well, there still is plenty of activity under sea ice. Therefore the more precise description should be 'reduces...'

Line 51-52: Reference 11 is used to manifest high T in NE Siberia during the Eemian....Well, the study referred to was actually carried out in NW-Central Siberia....

Figures:

Fig5: arrows indicating katabatic wind directions strike me and seem odd, especially for the ESCIS, where the flow is East to West!

Fig 6: These time-interval sections are fanciful and while such cartoons are often well received by the media, are they also realistic and founded on dependable data? I have my doubts...especially what the actual thickness of the assumed shelf ice sheet is concerned....Delete the whole figure and integrate essential interpretations into a new Fig. 5 composed of 2 panels: a) stage 6 with polynas; b) LIG with exisiting but partly open summer sea-ice cover over the AO basin....

Model: The model experiments leaves me somewhat confused, and I wonder about its validity in terms of near-realistic reproduction of other observations. While there is basically no difference between the time slices for March (including PI) LIG-120 has a lot of ice during June and September. That contradicts subarctic marine data, as well as Greenland ice which all indicate rather high temperatures exactly during this later phase of the Eemian.... (Landais et al 2016, Clim.Past; Govin et al. 2011, ClimPast).

This leads on to LIG sea level rise. Although rather debatable in terms of its timing globally, but it is clear that at some time, certainly still at 130 ka, sea level in the Arctic was likely lower, and not covering the rather wide Siberian shelves. But this has tremendous effects on the spatial size of the sea ice...

And, what about the opening of the Bering Strait? I think the modelling part needs to be really improved by introducing 2 more scenarios at 130 ka: one with still closed Bering Strait and non-flooded shelves; another with open Bering Strait and half-flooded shelves....

In the supplementary, I believe some of the data shown are without proper referencing (e.g., Fig. 3); Also, stable isotope data are shown for PS51/038-3, what about the other cores from the central AO (e.g. PS2200) aren't there any such data at all?

In summary, the topic has high potential, but judging the manuscript as is there is a need to have a much tighter focus on the core of the story.

Reviewer #3 (Remarks to the Author):

Stein et al. reconstruct past sea ice conditions from four sediment cores in the Arctic Ocean. They employ the relatively new IP25 method combined with a proxy for ice-free water to ameliorate the ambiguity of IP25 for situations with either permanent sea ice or total lack of ice. Their main results include qualitative reconstructions of sea ice ranging from the penultimate glacial period through the Last Interglacial (LIG) until the early part of the last glaciation (~150 to ~70 thousand years before present).

The results are novel and important. Although the discussion of the results is often speculative and vague, this is in my opinion acceptable here due to the novelty of the data that already represents a significant advance over the previous state-of-the-art. I think this paper is very well suited for Nature Communication and will receive significant attention as the field of sea ice reconstructions develops further and more data is obtained. The discussion of the implications of the new findings could be improved as outlined below. The integration of model results is scarcely motivated and seems like an afterthought. The quality of the figures is good. As a consequence, my criticism is minor and mostly concerns the connection with existing studies. As a caveat, it is important to note that I am not qualified to comment on the robustness or quality of the proxy data.

Given the likely substantial Arctic warming during the LIG (NEEM community members, 2013; Otto-Bliesner et al., 2013), I believe the result that sea ice conditions during MIS5e resembled those of today is quite unexpected. Since the contemporary decrease in sea ice is used repeatedly as a motivation (abstract, introduction), the manuscript should expand the discussion of the implications. If air temperatures really were much higher during MIS5e than today, one of the few remaining ways to maintain a sizeable sea ice cover is a reduced ocean heat transport. Such a change was found in two independent model simulations (Born et al., 2010, 2011). Sea surface temperatures in the Nordic Seas show that ocean heat input from the North Atlantic was higher during the early parts of MIS5d than during MIS5e, where the former has a orbital configuration very similar to today (Bauch et al., 2007, 2008; Risebrobakken et al., 2005, 2007).

The connection between the original proxy data and the previously published model simulations is not very clear. The purpose of including the model appears to be the confirmation of the proxy results, but since sea ice models are plagued by enormous uncertainties this argument is rather weak. It would be nice to have some more detail on the mechanisms of sea ice reduction in the MIS5e and the RCP scenarios. An energy balance calculation could help to understand why there is more sea ice during MIS5e than in the RCP scenarios and whether this is due to atmospheric or oceanic fluxes. Lastly, the consequences of a reduced sea ice cover should briefly be discussed, also in view of potentially too high terrestrial temperature reconstructions for MIS5e. A suitable recent reference is Merz et al. (2016).

Minor comments:

Some language editing will be necessary.

line 62: prEjections

line 79: reconstructionS

line 126 and throughout the manuscript: "saison" and "saisonal" should be "season" and "seasonal".

line 158: Please mention the model name here too, not just in the methods section.

line 172: "controverse" should read "controversal"

line 187: change "does" to "do" (plural)

line 195: Reference 62 is now somewhat outdated and should be replaced with Hughes et al. (2016)

line 202: "inflow" twice

line 208ff: This sentence is hard to understand. Please rephrase.

line 228ff: This paragraph is very speculative and should be noted as such. For example, it is not clear why the ice shelf should retreat during MIS6.

line 295ff: Without further detail it is not clear whether this agreement is coincidence.

line 324: "aware about" please rephrase

reference 29: publication year is missing

figure 4: excessive horizontal gaps between subfigures could be reduced to the same spacing as in the vertical.

figure 8: Please add a description of the horizontal orange bands in the figure or the figure caption.

#####

References:

Otto-Bliesner et al. (2013), How warm was the last interglacial? New model–data comparisons, *Philos. T. R. Soc. Lond.*, 371, 20130097, doi:10.1098/rsta.2013.0097

Bauch, H. A., and H. Erlenkeuser (2008), A "critical" climatic evaluation of last interglacial (MIS 5e) records from the Norwegian Sea, *Polar Res.*, 27, 135–151.

Bauch, H. A., and E. S. Kandiano (2007), Evidence for early warming and cooling in North Atlantic surface waters during the last interglacial, *Paleoceanography*, 22, PA1201, doi:10.1029/2005PA001252.

Risebrobakken et al. (2005), The extent and variability of the Meridional Atlantic Circulation in the Nordic Seas during Marine Isotope Stage 5 and its influence on the inception of the last glacial, in *The Nordic Seas: An Integrated Perspective*, *Geophys. Monogr. Ser.*, vol. 158, edited by H. Drange et al., pp. 323–339, AGU, Washington, D. C.

Risebrobakken et al. (2007), Inception of the Northern European ice sheet due to contrasting ocean and insolation forcing, *Quat. Res.*, 67, 128–135.

Merz et al. (2016), Warm Greenland during the last interglacial: the role of regional changes in sea ice cover
Climate of the Past 12, 2011–2031

Born et al. (2011), Late Eemian warming in the Nordic Seas as seen in proxy data and climate models

Paleoceanography 26, PA2207

Born et al. (2010), Sea ice induced changes in ocean circulation during the Eemian
Climate Dynamics 35(7-8), 1361-1371

Hughes et al. (2016), The last Eurasian Ice Sheets - a chronological database and time-slice
reconstruction, DATED-1. Boreas, 45, 1-45, doi:10.1111/bor.12142

Point-by-point response to the referees' comments

(1) Comments of Reviewer #1 (in italics) and our response

***Reviewer 1:** "...The study includes highly important findings based on the biomarker data and supported by microfossil data from previous work..... This work is definitely of interest to a wide audience in several fields and of an importance and novelty that warrants publication in Nature Communications*"

Authors: Thanks for this very positive general statement!!

***Reviewer 1:** "... , namely adding – if at all possible – data on HBI-III. I suspect the authors will have analysed this biomarker alongside IP25 Collins et al 2013. Belt et al. 2015... strong indications it could be used as a marginal ice zone (MIZ) / polynya indicator..... use of such a proxy would certainly further strengthen the interpretations made..... "*

Authors: We are very grateful for this important comment. We have checked our biomarker records for HBI-III. In the central Arctic Ocean cores PS2200 and PS51/38 characterized by the more or less absence of phytoplankton biomarkers, also HBI-III is absent. Limited but still significant amounts of HBI-III were found in the lower part of Core PS2757. In core PS2138, on the other hand, HBI-III occurs contemporaneously with the other phytoplankton biomarkers throughout. The HBI-III records allow a much more detailed interpretation of the data and strengthen our sea ice history. We also discuss our biomarker data in relationship to the studies of Collins et al. (2013) and Belt et al. (2015) as recommended by Reviewer 1.

Reviewer 1 also found a large number of smaller mistakes/typos in the text (marked in the pdf attached to the review) that we have considered/corrected in the revised version of the manuscript. Some of the comments/points are outlined in the following:

***Reviewer 1:** Line 81 (first version of the manuscript) – change "ostracodes" into "ostracods"*

Authors: We have not followed this suggestion as both writings are correct. We followed Cronin et al. (2010, 2013), authors of the papers cited in our manuscript, who use "ostracodes".

***Reviewer 1:** "While there is a slight reduction of the sea ice coverage in the early LIG (130 ka) and mid LIG (125 ka) time slices during summer (June), the most pronounced response is seen during the late autumn (September) where both early LIG (130 ka) and mid LIG (125 ka) display significant*

coverage reduction relative to PI“ -> rephrase this sentence.

Authors: Sentence has been rewritten as suggested by the reviewer (revised manuscript, lines 160-164)

Reviewer 1: *Line 279 (first version of the manuscript) – change “dinocysts“ into “dinoflagellate cysts“*

Authors: We have used now “dinoflagellate cysts“ throughout the revised manuscript.

Reviewer 1: *Lines 279/280 (first version of the manuscript) – revise interpretation of the occurrence of dinoflagellate species “Impagidinium pallidum“*

Authors: We have clarified the interpretation of the occurrence of dinoflagellate species “Impagidinium pallidum“, following the authors Matthiessen and Knies (2001) (revised manuscript, lines 286-289).

Reviewer 1: *Lines 309-310 (first version of the manuscript) – “We furthermore examine possible sea ice concentrations in future projection scenarios for RCP4.5 and RCP6.“ -> authors should add more discussion on boundary conditions during the Eemian vs. the near-future and possible reasons for the large difference in runs for the LIG-130/LIG-125 vs. RCP4.5 and RCP6*

Authors: We have extended the discussion of the LIG-130/LIG-125 vs. RCP4.5 and RCP6 scenarios significantly (revised manuscript, lines 366-377).

Reviewer 1: *Figure 1 - Can't see the green line well enough (at least not in the printout. Please use another colour.*

Authors: Colour has been changed (see new Fig. 1).

Reviewer 1: *Figure 3 – „seasonal“ misspelled*

Authors: The mistake has been corrected in the figure as well as in the main text

(2) Comments of Reviewer #2 (in italics) and our response

Reviewer 2: *“..... Overall, this is an interesting topic and some of the data are indeed quite unique/new by comparison to previous studies from the Arctic Ocean. Also, the main messages contained in the abstract are timely, thus, the story has potential to be published in Nat Comm eventually. Having said, the implications of the interpretation of the proxy data remains somewhat shallow, in particular, there is little use being made of the model results leaving the entire story rather underdeveloped is not really focussed on those 2 critical messages talked about in the abstract..... the manuscript has a strong review character reiterating stratigraphic and other data that have been discussed and published at length elsewhere..... The discussion part remains by and large descriptive..... “*

Authors: Thanks for this very positive, general statement. Concerning the interpretation of the proxy data, especially also in combination with the modelling results, the discussion has been enlarged significantly goes much more in depth now (see lines 295-377). Here, we have concentrated on the

two main messages of the paper, i.e., the polynya-like conditions off major circum Arctic ice sheets during MIS 6 and the existence of sea ice in the central Arctic Ocean during the LIG/Eemian, a climate interval warmer than today. We do not agree that our paper has more a review character. As stated very clearly, with our data for the first time semi-quantitative sea ice data of the central Arctic Ocean are presented. These data are then discussed in context with existing other (more qualitative) data. Of course, a stratigraphic framework precise enough to identify the MIS 6/MIS 5 time interval is needed. Here, we just present for each core one figure in the supplementary part to demonstrate for the reader that our age control is good enough to „sell our sea ice story“, but we did not describe in any detail the stratigraphies of our studied cores. Here references where the details can be found, are listed.

Reviewer 2: *“Line 51-52: Reference 11 (Kienast et al. 2011) is used to manifest high T in NE Siberia during the Eemian.... Well, the study referred to was actually carried out in NW-Central Siberia. “*

Authors: Here, we do not agree with Reviewer 2. We have used the term „NE Siberia“ as done by Kienast et al. (2011) in their original paper as well: Page 2135, Regional Setting, first paragraph - “The sampled exposure is situated in the coastal sector Oyogos Yar (72.68°N; 143.53°E) of the Yana-Indigirka lowlands between Cape Svyatoy Nos and the Merkushina Strelka Peninsula on the shore of Dmitry Laptev Strait in NE Siberia“.

Reviewer 2: *“Fig5 - arrows indicating katabatic wind directions strike me and seem odd, especially for the ESCIS, where the flow is East to West! “*

Authors: The direction of the arrows in the old Fig. 5 was misleading, sorry for this. It had nothing to do with real wind direction, it’s just a cartoon that should demonstrate katabatic winds related to ice sheets. We have revised the figure.

Reviewer 2: *“Fig 6 - These time-interval sections are fanciful and while such cartoons are often well received by the media, are they also realistic and founded on dependable data? I have my doubts...especially what the actual thickness of the assumed shelf ice sheet is concerned....Delete the whole figure and integrate essential interpretations into a new Fig. 5 composed of 2 panels: a) stage 6 with polynas; b) LIG with existing but partly open summer sea-ice cover over the AO basin.... “*

Authors: Figure 6 is a cartoon that displays the interpretation of the temporal succession of changes in sea ice and ice sheet at the Siberian continental margin, as demonstrated in the proxy records of Fig. 2c, in a very simple and convincing way. Thus, we think that this figure is very helpful and important for presenting our sea ice story to the reader, and we would like to have the figure included in the final version of our paper. However, we agree that the shelf ice sheet was not drawn very realistically in the old version; the figure has been revised accordingly.

Reviewer 2: *“..... While there is basically no difference between the time slices for March (including PI) LIG-120 has a lot of ice during June and September. That contradicts subarctic marine data, as well as Greenland ice which all indicate rather high temperatures exactly during this later phase of the Eemian.... (Landais et al 2016, Clim.Past; Govin et al. 2011, ClimPast). “*

Authors: We have discussed more clearly inconsistencies between model and proxy data, including results of Govin et al. (2011).

Reviewer 2: *“..... modelling part needs to be really improved by introducing 2 more scenarios at 130 ka: one with still closed Bering Strait and non- flooded shelves; another with open Bering Strait and half-flooded shelves.. “*

Authors: As Bering Strait was already re-opened during all the three LIG intervals and most parts of the shallow Siberian marginal seas were already flooded even during LIG-130 (Hu et al., 2010), we did not run additional scenarios with a closed Bering Strait and/or subaerial shelf seas here.

Reviewer 2: “.... I believe some of the data shown are without proper referencing (e.g., Fig. 3)..... “

Authors: We think that we have used the proper NC style of referencing, e.g., in Fig. 3: Fram Strait (FS)³⁶ and Early Holocene (EH) and Middle-Late Holocene (MLH) data from Fram Strait Core MSM5/5-712 (ref. 32).....

Reviewer 2: “.... stable isotope data are shown for PS51/038-3, what about the other cores from the central AO (e.g. PS2200) aren't there any such data at all? “

Authors: Oxygen and carbon isotope records of Core PS2200 have been added in Supplementary Figure 2.

(3) Comments of Reviewer #3 (in italics) and our response

Reviewer 3: . “..... The results are novel and important. Although the discussion of the results is often speculative and vague, this is in my opinion acceptable here due to the novelty of the data that already represents a significant advance over the previous state-of-the-art. I think this paper is very well suited for Nature Communication and will receive significant attention as the field of sea ice reconstructions develops further and more data is obtained. The discussion of the implications of the new findings could be improved as outlined below “

Authors: Thanks for this very positive general statement.

Reviewer 3: “..... The integration of model results is scarcely motivated and seems like an afterthought. The connection between the original proxy data and the previously published model simulations is not very clear. The purpose of including the model appears to be the confirmation of the proxy results, but since sea ice models are plagued by enormous uncertainties this argument is rather weak..... “

Authors: We are in line with this statement. Thus, we have extended the part of the discussion dealing with the model/proxy data significantly (see lines 324-380 of the revised manuscript).

Reviewer 3: “ If air temperatures really were much higher during MIS5e than today, one of the few remaining ways to maintain a sizeable sea ice cover is a reduced ocean heat transport. Such a change was found in two independent model simulations (Born et al., 2010, 2011). Sea surface temperatures in the Nordic Seas show that ocean heat input from the North Atlantic was higher during the early parts of MIS5d than during MIS5e, where the former has a orbital configuration very similar to today (Bauch et al., 2007, 2008; Risebrobakken et al., 2005, 2007). “.

Authors: We have extended the discussion, considering the results by Risebrobakken et al. (2005, 2007), Bauch and Erlenkeuser (2008), and Born et al. (2010) (see lines 295-338 of the revised manuscript).

Reviewer 3: “ *It would be nice to have some more detail on the mechanisms of sea ice reduction in the MIS5e and the RCP scenarios. An energy balance calculation could help to understand why there is more sea ice during MIS5e than in the RCP scenarios and whether this is due to atmospheric or oceanic fluxes.....* “.

Authors: An energy balance has been calculated, mechanisms of sea ice reduction have been discussed (see lines 295-338 of the revised manuscript and new Supplementary Fig. 9).

Reviewer 3: *Minor comments/mistakes in lines 62, 79, 126, 158, 172, 187, 195, 202, 208 of the first version of the manuscript.*

Authors: All mistakes/typos have been corrected in the revised version of the manuscript.

Reviewer 3: *Figure 4 - excessive horizontal gaps between subfigures could be reduced to the same spacing as in the vertical.*

Authors: Figure has been revised following the suggestion of the reviewer (see new Fig. 4).

Reviewer 3: *Figure 8 - Please add a description of the horizontal orange bands in the figure or the figure caption.*

Authors: Figure caption has been revised following the suggestion of the reviewer (see caption of Figure 8, lines 826-827).

Reviewer 3: *Add some important references in discussion of own data.*

Authors: As proposed by the reviewer, we have added/considered the following references: Otto-Bliesner et al. (2013), Bauch and Erlenkeuser (2008), Risebrobakken et al. (2005), Risebrobakken et al. (2007), Merz et al. (2016), Born et al. (2010), and Hughes et al. (2016).

Reviewers' comments:

Reviewer #1 (Remarks to the Author):

Although I liked the manuscript the first time I reviewed it, it has clearly improved due to revision. The authors have significantly extended the modelling section of the manuscript, which now is not "just an afterthought" as Reviewer 3 put it, but strengthens the overall story.

The authors have also included the biomarker HBI-III, which I asked them to consider. This has further strengthened the paper, as also the authors themselves state in their response letter. The authors have made the majority of corrections I asked for, certainly the more important ones (and seem to have done so also regarding the other reviewers' comments). Where they have not agreed, good reasons have been provided. Overall, the manuscript forms an even more coherent story, including very important findings as already stated in my previous report 1) During MIS 6, polynya-like conditions likely occurred off the major circum-Arctic ice sheets, contradicting a previously hypothesised giant MIS 6 ice shelf that would have covered the entire Arctic Ocean. 2) Even during the Eemian, when climate was clearly warmer than today, sea ice existed in the central Arctic Ocean during summer, whereas ice was significantly reduced along the Barents Sea continental margin influenced by Atlantic Water inflow.

I have again made a few (minor) corrections directly onto the manuscript.

I strongly recommend that this revised manuscript would be published in Nature Communications.

Reviewer #2 (Remarks to the Author):

I have read the revised version of the Stein et al contribution to Natcomm. I also had a closer look at the rebuttal letter by the authors, including their responses to the other reviewers. There is certainly an improvement seen in the revision. Although not very much going into further depth – improvements made by and large remain basically superficial as none of the more critical larger changes, such as the model as well figures.... – the essence of the manuscript is still a worthwhile contribution and should be published.

Having said that, there is one issue that still disturbs me, namely the somewhat loose treatment of the important fact of post-glacial sea-level rise after Stage 6 in combination with the opening of the Bering Strait. No doubt, global sea level rise during Termination 2 is very much debated, and for that reason we basically don't know nothing! However, the authors somewhat neglect that point by simply referring now to a model study by Hu et al – the study had nothing to do with a sea level rise reconstruction per se – to claim that the Bering Strait was already open by 130 ka. This I consider inadequate referencing, and I suggest to the authors to rethink how to solve or better how circumvent the fact that they do not want to provide model simulations for an open and closed strait respectively.

Reviewer #3 (Remarks to the Author):

l 322, incorrect citation: It is not the overturning circulation that weakened in the simulations of ref. 20 but the heat transport by the subpolar gyre. This is an important detail as the gyre has been found to be highly variable/vulnerable both in recent observations and several proxy records.

There are several problems with Supplementary figure 9:

- 1) The main text argues that the RCP simulations have a different seasonality as the LIG simulations due to the nature of CO₂ vs. orbital forcing (l. 375ff). This is not reflected by the annual average picture in Supp. figure 9.
- 2) The anomalies of the surface downwelling shortwave radiation at the bottom of the atmosphere (surface) are much larger than at the top of the atmosphere. The only physical explanation are changes in cloudiness, which are not discussed. However, the fields in columns a) and b) looks so very different that I suspect a mistake.
- 3) Column c) basically only reflects the consequences of the changes in sea ice cover, not the causes. I made my original comment to better understand the role of ocean heat transport, about which no inferences can be made based on this analysis.
- 4) The caption states that the units are Wm² instead of W/m² or Wm⁻². Column c) is labelled "Total Energy" but should read "Total Energy flux".

In summary, I do not think that this figure in its current form adds value to the manuscript and it should therefore either be corrected or removed.

Point-by-point response to the referees' comments

(1) Comments of Reviewer #1 (in italics) and our response

***Reviewer 1:** “..... manuscript has clearly improved due to revision. The authors have significantly extended the modelling section strengthens the overall story. biomarker HBI-III has further strengthened the paper.....authors have made the majority of corrections Where they have not agreed, good reasons have been provided. Overall, the manuscript forms an even more coherent story..... I have again made a few (minor) corrections directly onto the manuscript.I strongly recommend that this revised manuscript would be published in Nature Communications.”*

Authors: We thank Reviewer 1 for this very positive statement!!
In our second revision, we have considered all the minor corrections/suggestions, highlighted in yellow (revised manuscript, line 45, 93/94, 109/110, 144, 245, 286/287, 348/349, 352/353, 358/359, 479/480, 495, 502, 507/508, 793, 824, 826, 827, 841).

(2) Comments of Reviewer #2 (in italics) and our response

***Reviewer 2:** “..... certainly an improvement seen in the revision. Although not very much going into further depth – the essence of the manuscript is still a worthwhile contribution and should be published.”*

Authors: We thank Reviewer 2 for this dominantly general positive statement!!

***Reviewer 2:** “ one issue that still disturbs me somewhat loose treatment of the important fact of post-glacial sea-level rise after Stage 6 in combination with the opening of the Bering Strait global sea level rise during Termination 2 is very much debated, authors somewhat neglect that point by simply referring now to a model study by Hu et al provide model simulations for an open and closed strait respectively.”*

Authors: Thanks for this important comment. Following the suggestion of Reviewer 2, we have carried out an additional model run simulating a LIG-130 scenario with a closed Bering Strait and half-flooded Siberian shelf seas, and the results are shown in a new Supplementary Figure 9 and discussed in lines 365 to 375. Furthermore, a graph with modelled monthly sea ice concentrations for this 130 ka scenario has also been included in Figure 8 of the main paper.

(3) Comments of Reviewer #3 (in italics) and our response

Reviewer 3: “ incorrect citation: It is not the overturning circulation that weakened in the simulations of ref. 20 but the heat transport by the subpolar gyre. “

Authors: We have corrected this point (Line 321).

Reviewer 3: “..... several problems with Supplementary Figure 9:
..... main text argues that the RCP simulations have a different seasonality as the LIG simulations due to the nature of CO₂ vs. orbital forcing not reflected in figurebasically only reflects the consequences of the changes in sea ice cover, not the causes. I do not think that this figure in its current form adds value to the manuscript and it should therefore either be corrected or removed.“

Authors: We followed the suggestion of Reviewer 3 and have decided to remove the old Supplementary Figure 9.

REVIEWERS' COMMENTS:

Reviewer #2 (Remarks to the Author):

Overall I find the manuscript by Stein et al. to be improved over the previous 2 versions, especially because the authors now considered a closed Bering Strait in the modelling effort. What still leaves me head-scratching somewhat is the Siberian shelf ice sheet during stage 6. First, there is no further proof of evidence given in this very ms that it existed to the extent as shown in Fig. 5 – not to mention the Second, if it had existed to the thickness as indicated in Fig. 6, was it also responsible for feeding a pan-arctic ice shelf cover with a thickness of a 1000m or more? I don't believe it...and that for a number of reasons. To elaborate on them would go well beyond the scope of this review – the entire issue of the Siberian shelf ice sheet is actually a topic on its own. But I just want to remind the authors about the definition of an ice shelf, which is a floating extension of an ice sheet that has formed on land from many thousands of years of snowfall forming glaciers which flow down to sea level by gravity... etc. Therefore, I strongly recommend to delete the, in my view, rather controversial and misleading panel 1 of Fig. 6. Once done, the paper should be published.

Point-by-point response to the referees' comments

Comments of Reviewer #2 (in italics) and our response

Reviewer 2: *“Overall I find the manuscript by Stein et al. to be improved over the previous 2 versions, especially because the authors now considered a closed Bering Strait in the modelling effort. What still leaves me head-scratching somewhat is the Siberian shelf ice sheet during stage 6. First, there is no further proof of evidence given in this very ms that it existed to the extent as shown in Fig. 5.....”*

Authors: We thank Reviewer 2 for this dominantly general positive statement!! Concerning the East Siberian Shelf (ESS) Ice Sheet, however, we think that we have strong evidence as also outlined in our manuscript. In Supplementary Figure 7 streamlined glacial landforms indicative of ice-shelf grounding at the southern end of the Lomonosov Ridge are put into stratigraphic context by sub-bottom data and a dated sediment core. These data make a strong point for both ice thickness of about 1000 m during MIS 6 over the Lomonosov Ridge and its source region on the ESS. We are convinced it is better to leave panel 1 of Figure 6 and discuss it rather than removing it and thereby ignoring our own evidence (see Stein et al., 2016).

Reviewer 2: *“.....not to mention the Second, if it had existed to the thickness as indicated in Fig. 6, was it also responsible for feeding a pan-arctic ice shelf cover with a thickness of a 1000m or more? I don't believe it...and that for a number of reasons. To elaborate on them would go well beyond the scope of this review – the entire issue of the Siberian shelf ice sheet is actually a topic on its own.”*

Authors: The set of evidence outlined above is in line with the interpretation of Jakobsson et al. (2016), although we agree that an ocean-wide coverage of a thick ice shelf cannot be deduced from the data presented in the manuscript. In order to accommodate this point we have added a question mark to the ice extent in panel 1 of Figure 6 and mention this in the text.

Reviewer 2: *“Therefore, I strongly recommend to delete the, in my view, rather controversial and misleading panel 1 of Fig. 6.”*

Authors: We disagree that the panel 1 is misleading, because it illustrates the evidence found in the sedimentary record of the southern Lomonosov Ridge and described in the paper.